# Hypothalamus-hippocampus circuitry regulates impulsivity via melanin-concentrating hormone

Emily E. Noble[1,2], Zhuo Wang[3], Clarissa M. Liu[4], Elizabeth A. Davis[1], Andrea N. Suarez[1], Lauren M. Stein[5], Linda Tsan[4], Sarah J. Terrill[1], Ted M. Hsu[6], A-Hyun Jung[4], Lauren M. Raycraft[7], Joel D. Hahn[8], Martin Darvas[9], Alyssa M. Cortella[1], Lindsey A. Schier[1,4], Alexander W. Johnson[7], Matthew R. Hayes[5], Daniel P. Holschneider[3] & Scott E. Kanoski[1,4]*

Behavioral impulsivity is common in various psychiatric and metabolic disorders. Here we identify a hypothalamus to telencephalon neural pathway for regulating impulsivity involving communication from melanin-concentrating hormone (MCH)-expressing lateral hypothalamic neurons to the ventral hippocampus subregion (vHP). Results show that both site-specific upregulation (pharmacological or chemogenetic) and chronic downregulation (RNA interference) of MCH communication to the vHP increases impulsive responding in rats, indicating that perturbing this system in either direction elevates impulsivity. Furthermore, these effects are not secondary to either impaired timing accuracy, altered activity, or increased food motivation, consistent with a specific role for vHP MCH signaling in the regulation of impulse control. Results from additional functional connectivity and neural pathway tracing analyses implicate the nucleus accumbens as a putative downstream target of vHP MCH1 receptor-expressing neurons. Collectively, these data reveal a specific neural circuit that regulates impulsivity and provide evidence of a novel function for MCH on behavior.

---

[1] Human and Evolutionary Biology Section, Department of Biological Sciences, University of Southern California, Los Angeles, CA 90089, USA. [2] Department of Foods and Nutrition, University of Georgia, Athens, GA 30606, USA. [3] Department of Psychiatry & Behavioral Sciences, University of Southern California, Los Angeles, CA 90089, USA. [4] Neuroscience Graduate Program, University of Southern California, Los Angeles, CA 90089, USA. [5] Department of Psychiatry, Perelman School of Medicine, University of Pennsylvania, Philadelphia, PA 19104, USA. [6] Department of Psychology, University of Illinois at Chicago, Chicago, IL 60612, USA. [7] Department of Psychology and Neuroscience Program, Michigan State University, East Lansing, MI 48824, USA. [8] Neurobiology Section, Department of Biological Sciences, University of Southern California, Los Angeles, CA 90089, USA. [9] Department of Pathology, University of Washington, Seattle, WA 98195, USA. *email: kanoski@usc.edu

mpulsivity, or responding without apparent forethought for the consequences of one's actions, is associated with several psychiatric disorders, including drug addiction, excessive gambling, affective disorders, attention-deficit hyperactivity disorder, and Parkinson's disease[1]. Recent attention has also implicated a relationship between behavioral impulsivity and excessive food intake[2,3], binge eating disorder[4], weight gain[5], and obesity[6,7]. Although impulsivity per se is not always maladaptive, it can often lead to consequences that are undesired or unintended by the individual. Understanding the neural substrates regulating impulsivity may lead to the development of novel treatments that can improve quality of life for individuals struggling with disorders involving excessive behavioral impulsivity.

Behavioral impulsivity can be divided into two distinct categories: impulsive action and impulsive choice[8]. Impulsive action refers to a failure to inhibit an inappropriate response to a stimulus, whereas impulsive choice is characterized by impulsive decision-making caused by a distorted and/or poor consideration of future behavioral consequences, often evaluated experimentally as the preference of a small immediate reward over a larger delayed one. Reliable rodent models have been established to study both impulsive action and impulsive choice, and the literature supports a strong neurobiological and behavioral homogeneity in these measures across species[9,10]. While a comprehensive understanding of the neural circuitry governing both aspects of impulsivity is lacking, some evidence indicates that impulsive action and impulsive choice have both common and distinct underlying neural substrates[11].

Recent findings link the ventral hippocampus (vHP) in the control of impulsive and motivated responding for palatable food[12]. This brain region, more classically involved in learning and memory, responds to a multitude of interoceptive signals (endocrine, neuropeptidergic) to regulate learned and inhibitory aspects of food intake and other motivated behaviors[13]. A possible source of interoceptive signaling to vHP neurons is the lateral hypothalamic area (LHA), a multi-region hypothalamic division involved in the control of fundamental behavioral and physiological processes[14–16]. Regional organization of the LHA is informed by the expression of various signaling molecules. One such signal is melanin-concentrating hormone (MCH), a 19-amino acid peptide that is synthesized predominantly in LHA and zona incerta (ZI) neurons[17,18] and has been associated with both food- and drug-seeking behaviors[19–24]. The mammalian MCH receptor 1 (MCHR1) is expressed in vHP neurons[22,25]; however, the functional relevance of this system has not been explored. Here we investigate the role of vHP MCH signaling in the control of behavioral impulsivity. Our results show that MCH signaling in the vHP increases impulsive responding and impulsive choice for a palatable food reinforcer but has no effect on food-motivated responding, locomotor activity, or clock speed timing. We conclude that the projection pathway from MCH neurons in the LHA/ZI to the vHP plays a role in mediating impulsivity.

## Results

**Central MCH signaling increases food impulsivity.** To determine whether central MCH signaling affects impulsive action, adult male Sprague Dawley rats were tested in the differential reinforcement of low rates of responding (DRL) task. DRL measures the capacity for an animal to withhold lever pressing for a palatable high-fat/high sucrose food pellet for a predetermined period of time and is an established model of measuring impulsive action in rats[26]. Animals are first trained (without food restriction) to wait for 20 s between lever presses to receive a single 45 mg reward pellet (Fig. 1a, b). Lever pressing for a reinforcer prior to the end of the 20-s interval results in no reinforcement delivery and the 20-s interval time clock restarts. Intracerebroventricular (ICV) injection of 5 μg MCH resulted in an increased number of presses on the reinforced (active) lever with no significant differences in the number of pellets earned, resulting in an overall reduced efficiency in the task, indicating increased impulsivity (Fig. 1c–e). MCH had no effect, however, on lever pressing activity on the inactive (non-reinforced) lever (Student's two-tailed, paired $t$ test; means [SEM]: Vehicle 5.1 [1.2], MCH 4.5 [1.1]; $n = 10$). Consistent with previous reports[27,28], ICV injection of MCH also dose-dependently increased chow intake (Supplementary Fig. 1) in non-restricted animals free-feeding in the home cage.

To determine whether selective activation of MCH neurons similarly elevates impulsive responding in the DRL task, we utilized a virally mediated chemogenetics approach. Animals were injected with a custom AAV2-MCH DREADDs-hM3D(Gq)-mCherry (Fig. 1f; ref. [29]) targeting the LHA region with hM3D (Gq) "DREADDs" (designer receptors exclusively activated by designer drugs) expression under the control of an MCH gene promoter. Results from mCherry quantification showed that using this approach ~84% of MCH neurons were transduced and thus carried the excitatory DREADDs, and consistent with our previous report, the mCherry transgene was selectively expressed in MCH neurons[29]. Animals were trained in the DRL task until they reached an efficiency plateau (Fig. 1g). On test days, animals were injected with either ICV Clozapine-N-oxide (CNO; DREADDs ligand) or vehicle using a within-subjects, counterbalanced design. Similar to ICV injection of MCH, chemogenetic activation of MCH neurons produced a trend for increased number of active lever presses without significantly influencing the number of pellets earned in the DRL task, resulting in a significant reduction in efficiency compared to vehicle treatment (indicating increased impulsive responding for food (Fig. 1h–j). There were no differences between treatments in the number of presses on the inactive lever (Student's two-tailed, paired $t$ test; means [SEM]: Vehicle 4.7 [1.2], CNO 4.7 [1.4]; $n = 15$). Importantly, ICV CNO treatment had no effect on performance in the DRL task in rats with no hM3D(Gq) expressed in MCH neurons (Supplementary Fig. 2), suggesting that the results described above are not secondary to nonspecific effects of CNO[30].

**MCH signaling to CA1v modulates impulsive action and choice.** MCHR1 expression has been reported in the vHP[22,25] but the functional relevance of this system has not been previously examined. Immunohistochemistry (IHC) results confirmed a high density of MCH immunoreactive (MCH-ir) fibers throughout all vHP subregions (Supplementary Fig. 3). Here we targeted the ventral CA1 (CA1v) subregion of vHP as we have recently identified a role for CA1v in the control of motivated responding for food[12,31]. Fluorogold (FG) anatomical retrograde pathway tracing data showed that ~5% of MCH neurons colocalise with FG (Fig. 2a). We next revealed robust MCHR1 expression in the CA1v and further investigated the neuroanatomical phenotype of MCHR1-containing neurons in this subregion. Fluorescence in situ hybridization (FISH) data reveal that ~44% of MCHR1 containing neurons in the CA1v colocalize with the GABAergic marker GAD2, whereas ~83% of MCHR1-containing neurons colocalize with the glutamatergic marker vGLUT1 (Fig. 2b, c), indicating that the MCHR1 is expressed in both excitatory and inhibitory neurons with an overall predominance in the glutamatergic pyramidal layer of CA1v.

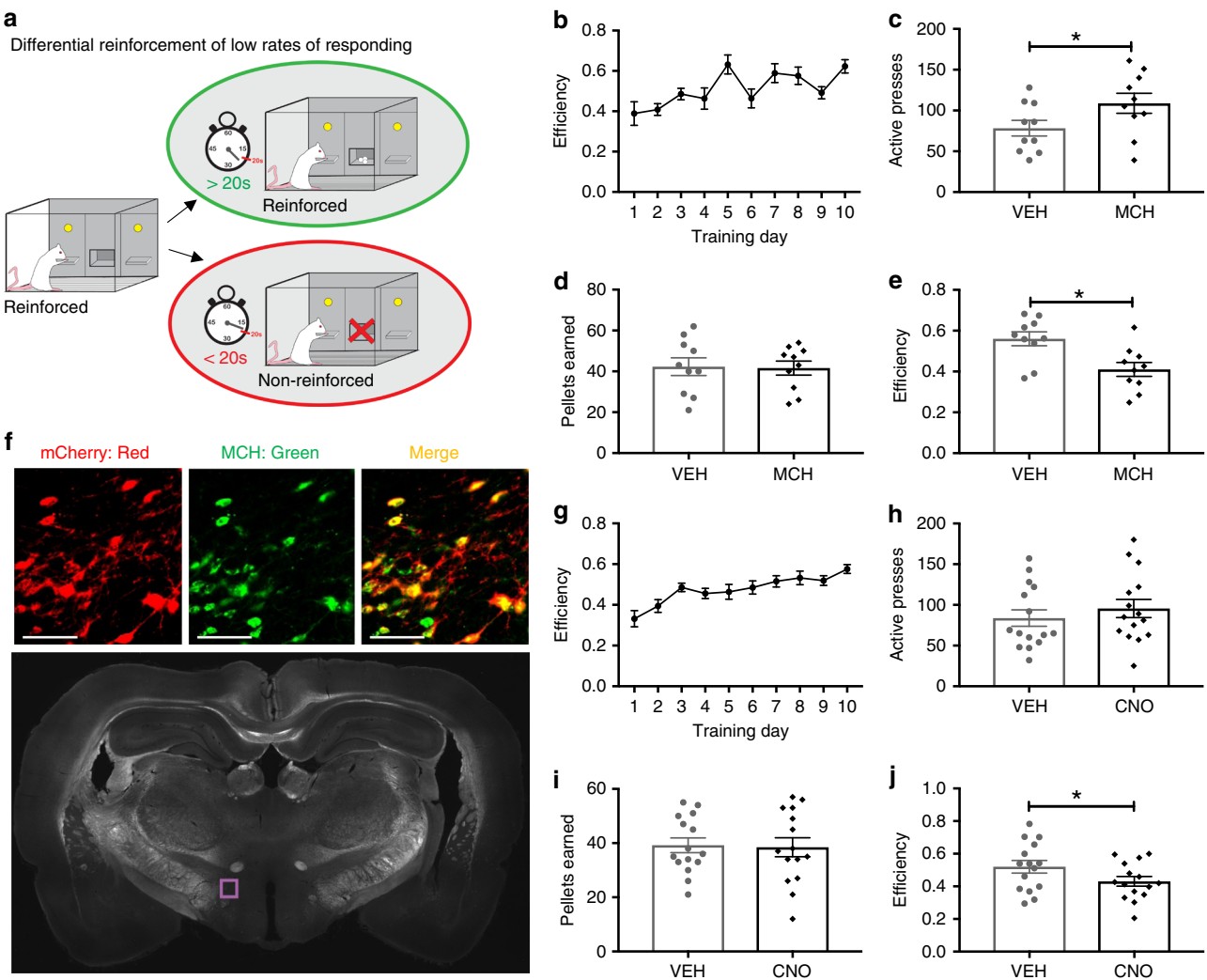

**Fig. 1** MCH signaling increases impulsive responding for food. **a** A schematic diagram of the differential reinforcement of low rates of responding task (DRL). **b–e** Intracerebroventricular (ICV) injection of MCH (5 μg) effects on food impulsivity in the DRL task: **b** efficiency data during the training phase under the DRL 20 schedule (**c–e**; data were analyzed using Student's two-tailed, paired $t$ tests; $n = 10$), **c** active lever presses in the DRL test session ($P = 0.006$), **d** number of pellets earned in the DRL test session, and **e** efficiency in the DRL test session ($P = 0.0009$). **f** Representative images showing the localization of the mCherry fluorescence reporter (red) in MCH neurons (green) following injections of AAV2-rMCHp-hM3D(Gq)-mCherry (colocalization in yellow). The region shown in the upper images is indicated by the purple box outlined in the coronal rat brain section below (~3.5 mm posterior to bregma). **g–j** Chemogenetic activation of MCH DREADDs-containing neurons following ICV CNO injections effects on food impulsivity in the DRL task. **g** Efficiency data during the training phase in DRL 20 (**h-j**; data were analyzed using Student's two-tailed, paired $t$ tests; $n = 15$), **h** number of active lever presses in the DRL test session, **i** number of pellets earned in the DRL test session, and **j** efficiency in the DRL test session ($P = 0.02$). Data shown as mean ± SEM; scale = 50 μm; *$P < 0.05$. Source data are provided as a source data file

We next tested whether the CA1v is a site of action for MCH effects on impulsive action. Direct pharmacological injection of 1 μg of MCH into the CA1v increased the number of active lever presses in the DRL task without increasing the number of pellets earned, thus yielding a significant reduction in efficiency relative to vehicle treatment (indicating greater impulsivity) (Fig. 3a–d). There were no differences between treatments, however, in presses on the inactive lever, (Student's two-tailed, paired $t$ tests; means [SEM]: Vehicle 2.5 [0.6], MCH 2.3 [0.6]; $n = 12$). We next tested whether chemogenetic activation of MCH neurons that project to the CA1v would also increase behavioral impulsivity. Unlike the pharmacological approach, chemogenetic activation of CA1v-projecting MCH neurons likely involves co-release of MCH with other neurotransmitter and neuropeptidergic signals expressed in MCH neurons, thereby better approximating

endogenous activation of the MCH → CA1v pathway. To achieve this, we utilized a dual virus chemogenetic approach to activate MCH neurons that specifically project to the CA1v[29,32]. The retrograde-transported canine adenovirus type 2 containing the cre recombinase transgene (CAV2-CRE)[32] was injected into the CA1v and a cre-dependent MCH-DREADDs adeno-associated virus (AAV) type 2 (AAV2-DIO-MCH DREADDs-hM3D(Gq)-mCherry) was injected into the LHA and ZI (Fig. 3e). With this approach, expression of the hM3D(Gq) DREADDs are restricted to MCH neurons that have axon terminals within the CA1v. By this technique (~23%) of MCH neurons that were targeted with non-CRE-dependent DREADDs approach (described above) were infected (Fig. 3f). ICV CNO injections in these animals revealed that, similar to CA1v-targeted MCH pharmacology, activation of CA1v-projecting MCH neurons did

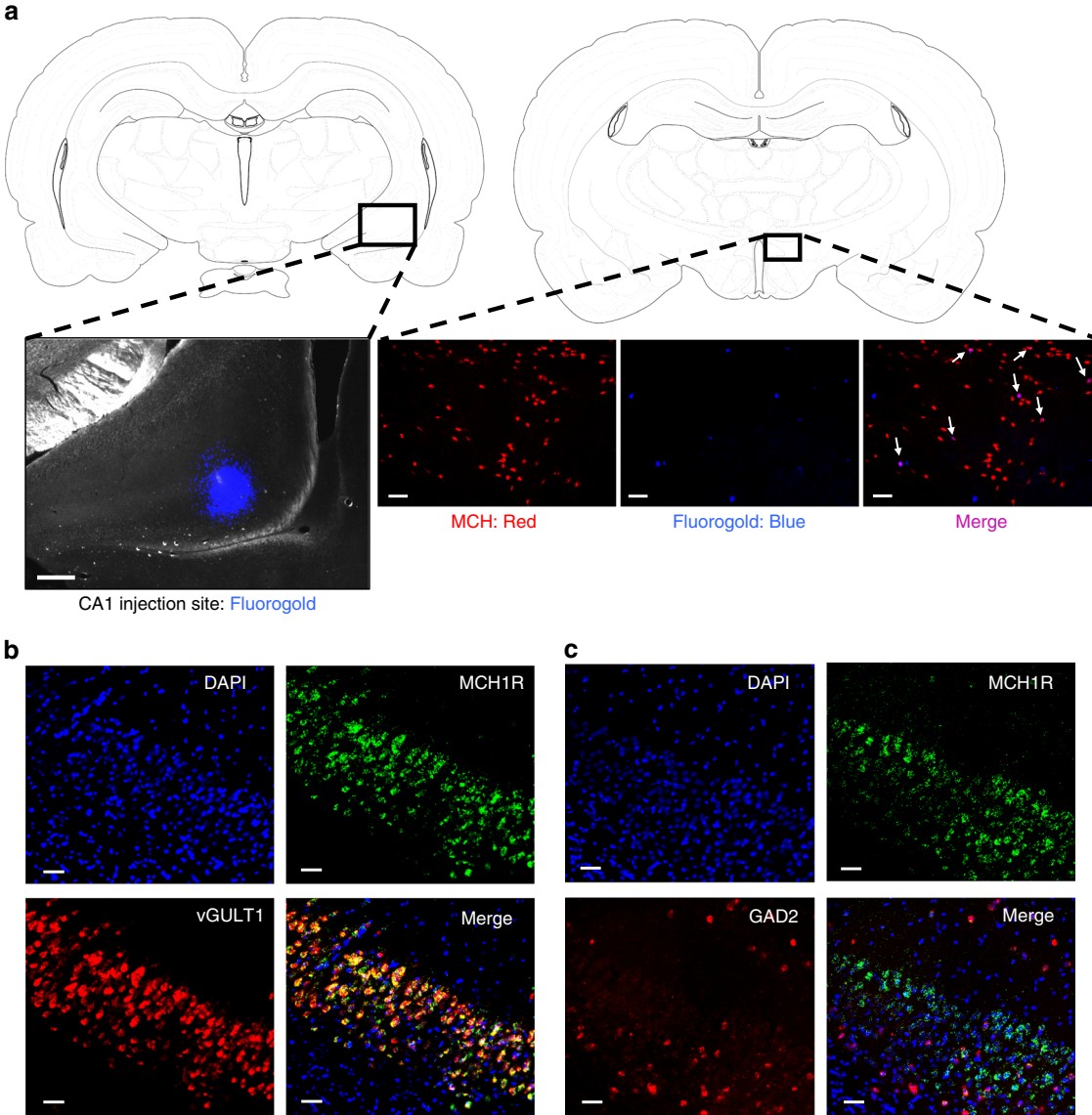

**Fig. 2** MCH neurons communicate to glutamatergic neurons in the CA1v. **a** (left) A representative injection site for fluorogold in the CA1 region of the vHP; scale = 200 μm. (right) Fluorogold (pseudocolored blue) colocalizes with MCH immunofluorescence (red) in a region of the lateral hypothalamic area (white arrows indicate colocalization); scale = 100 μm. **b** Fluorescence in situ hybridization for MCHR1 mRNA (green) and the glutamatergic marker vGLUT1 (red), with DAPI counterstain (blue). **c** Fluorescence in situ hybridization for MCHR1 mRNA (green) and the GABAergic marker GAD2 (red), with DAPI counterstain (blue); scale = 50 μm

not affect lever presses on the inactive lever (Student's two-tailed, paired *t* tests; means [SEM]: Vehicle 2.8 [0.7], CNO 4.5 [1.2]; *n* = 12) but increased the number of active lever presses without increasing the number of pellets earned, thus yielding reduced efficiency in the task (Fig. 3g–j).

To determine whether CA1v MCH effects on impulsive action extend to other impulsive behaviors, the delay discounting task was used to examine impulsive choice behavior. Delay discounting is characterized as the depreciation of the value of a reinforcer as the time it takes to obtain the reward is increased[33]. We hypothesized that MCH in the vHP would increase delay discounting behavior. During training, animals responded more on the immediate reinforcement lever during the last block of the task, when the time delay for the larger reinforcement increased to 45 s (Supplementary Fig. 4). During testing, there was a significant drug × block interaction, whereby CA1v injection of

MCH reduced the percentage of responding for the larger reward when the time delay was longer (30–45 s) relative to vehicle treatment, indicating an increase in impulsive choice behavior (Fig. 4a).

The DREADD-mediated activation of vHP-projecting MCH neurons approach described above may engage other brain regions via branching collateral projections of these neurons. To identify putative collateral projections of vHP-projecting MCH neurons, a conditional dual viral pathway-tracing approach was combined with IHC (described in Supplementary Methods). We observed evidence of collateral projections from vHP-projecting MCH neurons in the basolateral amygdala (BLA) but in no other regions, as evident by co-labeling of MCH and tdTomato (fluorescent reporter in cre-dependent anterograde tracing AAV) immunoreactivity in axons innervating the BLA (Supplementary Fig. 5).

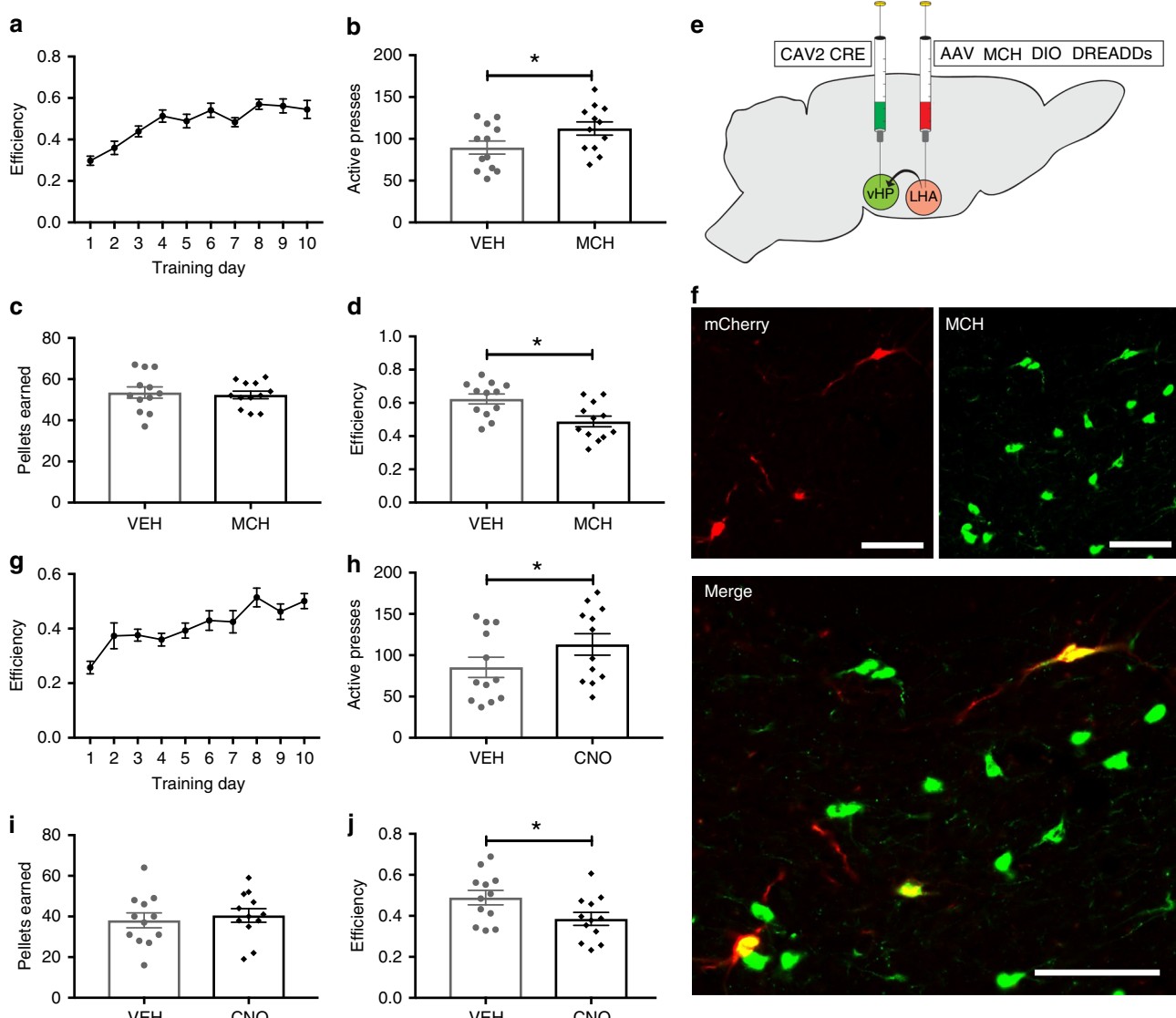

**Fig. 3** Activation of MCH-vHP signaling increases food impulsivity. **a–d** Effects of vHP injection of MCH on impulsive responding for food in the DRL task: **a** efficiency data during the training phase in DRL 20. (**b–d**; data were analyzed using Student's two-tailed, paired t tests; n = 12). **b** Number of active lever presses during test phase (P = 0.02). **c** Number of pellets earned during test phase. **d** Efficiency in the DRL test phase (P = 0.008). **e** Schematic cartoon depicting methods for dual virus chemogenetic approach. The retroactively transported canine adenovirus 2 (CAV2 CRE) was used to deliver the transgene for cre recombinase to vHP projection neurons following direct injection into the vHP. A cre-dependent adeno-associated virus containing excitatory MCH DREADDs–mCherry transgene (AAV2 MCH DIO DREADDs) was injected into the LHA and ZI. **f** Representative images of fluorescent reporter localization in MCH neurons. **g–j** Effects of chemogenetic activation of MCH neurons that project to the vHP on impulsive responding for food in the DRL test session: **g** efficiency data during the training phase in DRL 20. (**h–j**; data were analyzed using Student's two-tailed, paired t tests; n = 12), **h** number of active lever presses during DRL testing (P = 0.03), **i** number of pellets earned during DRL testing, and **j** efficiency during the DRL testing (P = 0.03). Data shown as mean ± SEM; scale = 50 μm; *P < 0.05. Source data are provided as a source data file

**CA1v MCH does not affect motivation, locomotion, or timing.** Given that both the DRL and delay discounting tasks involve motivated responding for food, as well as accurate timing capabilities, it is possible that apparent outcomes of CA1v MCH signaling on impulsivity are secondary to one or both of these effects. Further, both global MCH knockout[34–36] and pharmacological MCHR1 blockade[37] are associated with hyperactivity, suggesting that DRL and delay discounting results may be secondary to changes in physical activity. To investigate food-motivated behavior (independent of impulsivity), we first tested whether CA1v MCH signaling affects normal, home cage feeding behavior. Injection of 1 μg of MCH (a dose that increased both

impulsive action and choice) into the CA1v had no effect on chow intake in the animal's home cage (Fig. 4b), indicating that DRL effects are unlikely to be secondary to increased appetite/hunger. Also, consistent with these pharmacological data, ICV CNO-mediated activation of CA1v-projecting MCH neurons had no effect on home cage chow intake (Fig. 4c). Given that the impulsivity tasks utilize palatable high-fat/high-sucrose pellets, we next tested whether CA1v MCH signaling influences intake of palatable foods. Injection of 1 μg of MCH into the CA1v had no effect on palatable high-fat/high-sugar diet intake in the animal's home cage (Fig. 4d), indicating that DRL effects are unlikely to be secondary to increased appetite for palatable foods. We next

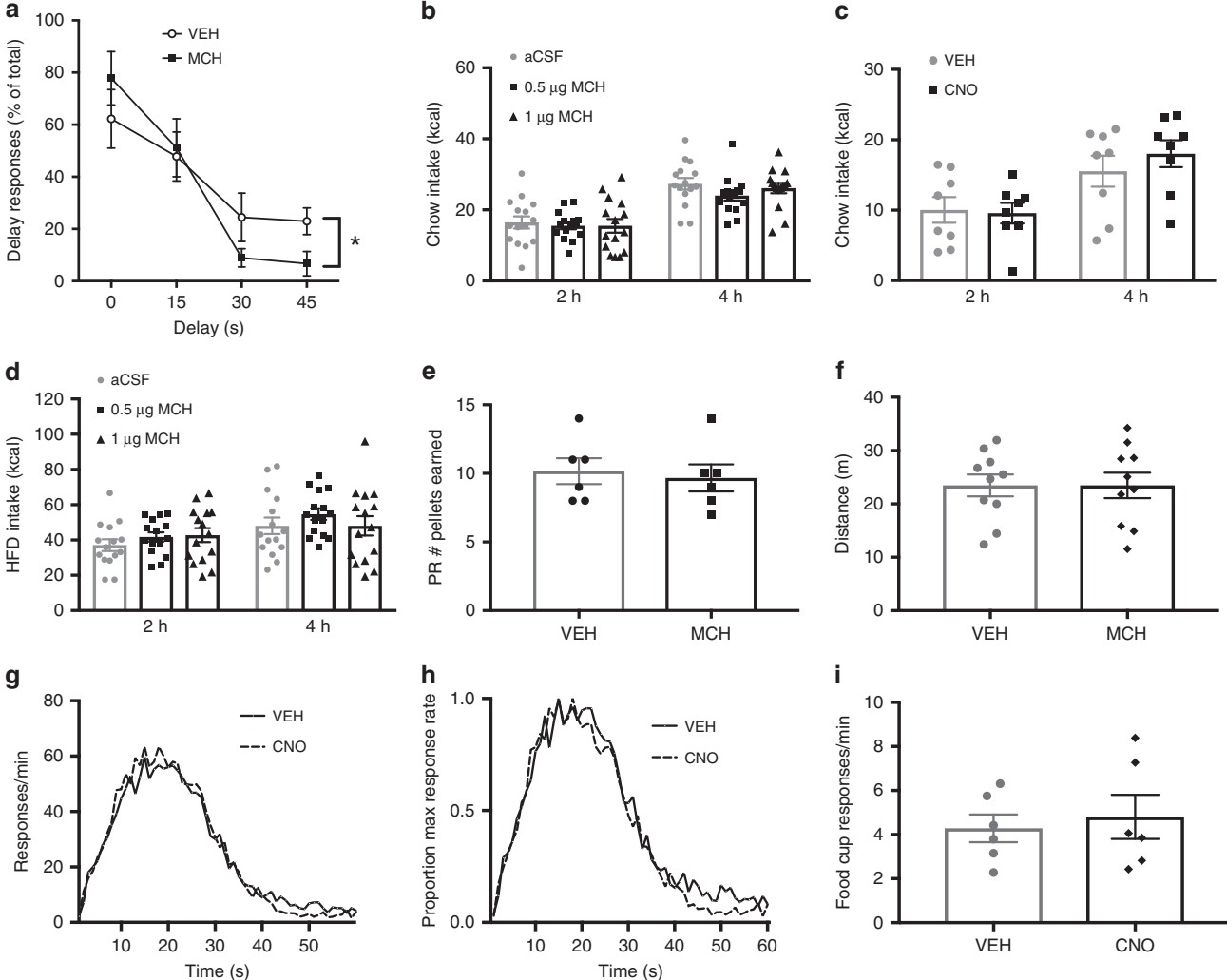

**Fig. 4** vHP MCH signaling increases impulsive choice but not food motivation or timing accuracy. **a** MCH (1 μg in the vHP) effects on performance in the delay discounting task (two-way repeated measures ANOVA with delay × drug treatment; $n = 9$). Results show a significant delay × drug interaction ($F_{(3,24)} = 5.53$; $P = 0.005$), with main effect of delay time ($F_{(3,24)} = 18.21$; $P < 0.0001$). (**b–d**; Two-way repeated measures ANOVA (time × drug)). **b** Effect of vHP MCH injection ($n = 15$) and **c** chemogenetic activation of vHP projecting MCH neurons ($n = 8$) on standard chow intake in the home cage. (**d** Effect of vHP MCH injection on high fat/high sugar (HFD) diet intake ($n = 15$). **e** MCH (1 μg in the vHP) effect on the number of pellets earned in the progressive ratio task (Student's two-tailed, paired $t$ test; $n = 6$). **f** Effect of 1 μg of MCH (0.5 μg/side) in the vHP on distance traveled in the open field test (Student's two-tailed, paired $t$ test; $n = 10$). **g–i** Effect of chemogenetic activation of MCH neurons on peak interval timing ($n = 6$). **g** Peak interval response rates during probe trials, with increased response rates that peaks near the 20 s criterion duration; **h** normalized function for peak rate, where the peak function for each animal was converted to a percentage of maximum response rate. This analysis revealed no horizontal shift in the peak interval timing function (two-way repeated measures ANOVA; $F_{(1,5)} = 1.63$, 0.25) following activation of MCH neurons by lateral ventricle CNO injections. **i** MCH neuronal activation also had no influence on food cup approach behavior during the peak interval test (Student's two-tailed, paired $t$ test). Data shown as mean ± SEM; (*$P < 0.05$). Source data are provided as a source data file

reasoned that the increase in delay discounting may be due to an increase in the incentive motivation to obtain the palatable food reward. We therefore tested the animals in the operant progressive ratio (PR) reinforcement schedule task, an established task for measuring motivated responding for the high-fat/high-sugar pellets[38]. However, we found that MCH in the CA1v had no effect on motivated responding in the PR task (Fig. 4e). Together these data suggest that the effects of CA1v MCH on impulsive behavior are unlikely to be driven by an increase in appetite, food palatability, or incentive motivation to obtain food. In addition, our data from the open field test reveal that, relative to vehicle treatment, 1 μg MCH deliver to the CA1v did not significantly influence distance traveled during the test session

(Fig. 4f), suggesting that the increased impulsivity outcomes are unlikely to be secondary to altered general levels of physical activity.

As timing is a component of both the DRL and delay discounting tasks, we next asked whether activating MCH neurons affects the animals' ability to accurately time 20 s using the peak interval timing task. We reasoned that, if vHP MCH were affecting internal clock timing, the animals would have a reduced efficiency in the DRL task not due to impulsivity but rather an inability to accurately time 20 s. Results from the peak interval task showed that chemogenetic activation of MCH neurons did not affect the animals' capacity to accurately time 20 s (Fig. 4g–l), as peak interval response rates during the probe

trials showed similar increased response rates near the 20 s criterion duration following both vehicle and CNO delivery (Fig. 4g). Additionally, when the peak function for each animal was converted to a percentage of maximum response rate, CNO did not influence the horizontal shift in the peak interval timing function (Fig. 4h). Importantly, MCH neuronal activation did not affect food cup approach behavior during the peak interval test (Fig. 4i), thus further corroborating that the impulsivity outcomes are not based on elevated food motivation. Overall, these findings show that activating MCH neurons did not affect the ability for the animals to accurately time 20 s and therefore reduced efficiency scores in the DRL task and reduced discounting in the delay discounting task are unlikely due to MCH influences on the ability to time events.

**Endogenous CA1v MCHR1 signaling regulates impulsive control**. Given that increasing MCHR1 activation in the CA1v increases impulsive responding for palatable food, we reasoned that virally mediated chronic MCHR1 mRNA knockdown using short hairpin RNA (shRNA) in the CA1v would reduce impulsive behavior. Site-specific injections of an AAV1-GFP-U6-r-MCHR1-shRNA reduced MCHR1 immunofluorescence and gene expression in the CA1v by ~67% (Fig. 5a, b, Supplementary Fig. 6a, b). Consistent with the results above, MCHR1 knockdown in the CA1v had no effect on food intake or body weight (Fig. 5c, d). However, contrary to our hypothesis, the DRL efficiency during the 20-s schedule was significantly reduced (indicating increased impulsivity) in rats with chronic CA1v MCHR1 knockdown (Fig. 5e–h), with no differences in lever pressing activity on the inactive lever (Student's two-tailed, unpaired $t$ test; means [SEM]: Control 7.4 [1.8], MCHR1 knockdown 6.3 [1.4]; $n$ = 11, 16).

To further confirm that loss of MCH signaling in the CA1v promotes elevated impulsive responding, we used a projection site-specific RNA interference (RNAi) approach. Following DRL 20 training, an additional cohort of rats were injected in the vHP with an AAV2 (retro)-GFP-U6-r-pMCH-shRNA, which is retrogradely transported to cells that project to the CA1v and drives expression of shRNAs targeting pMCH mRNA expression (Supplementary Fig. 6c, d; representative green fluorescent protein (GFP) expression in MCH neurons depicted in Supplementary Fig. 7). This shRNA approach achieved 94% knockdown of MCH mRNA in vitro, and a nonsignificant ~7% knockdown of global LHA MCH mRNA in vivo following CA1v AAV injections relative to controls, an effect consistent with our estimates that between 5% and 15% of all MCH neurons project to the CA1v (Figs. 2 and 3). Results show that animals that received pMCH RNAi in CA1v-projecting MCH neurons had similar body weight and food intake to animals injected with the control virus (Supplementary Fig. 8a, b). Animals with pMCH RNAi also showed no differences in active lever presses (Student's two-tailed, unpaired $t$ test; means [SEM]: CTL 47.3 [16.6], pMCH RNAi 47.7 [16.8]; $n$ = 8, 9), pellets earned (CTL 23.1 [5.9], pMCH RNAi 23.9 [6.5]), or efficiency scores (CTL 0.61 [0.06], pMCH RNAi 0.64 [0.06]) in the DRL task under non-food-restricted conditions; however, following 24-h food restriction, rats with RNAi targeting CA1v-projecting MCH neurons demonstrated increased activity on the reinforced lever during DRL 20 with no differences in the number of pellets earned, yielding significantly reduced efficiency (indicating increased impulsivity) in the task relative to controls (Supplementary Fig. 8c–f). Further, there were no differences in locomotor activity levels as measured by presses on the non-reinforced lever (Student's two-tailed, unpaired $t$ test; means [SEM]: Vehicle 4.5 [0.6], CNO 4.2 [1.5]; $n$ = 8, 9) or in distance traveled in the open field task (Supplemental Fig. 8g).

Taken together with the gain-of-function results above, these two chronic loss-of-function results suggest that CA1v MCH signaling bidirectionally regulates impulse control such that acute or chronic perturbations of MCH tone, in either direction, elevate impulsivity.

**The nucleus accumbens (ACB) is a candidate downstream target**. To determine brain regions engaged by CA1v MCH signaling, we mapped regional cerebral glucose uptake (rCGU) in awake animals at rest during a 45-min resting-state [$^{14}$C]-2-deoxy-D-glucose (2-DG) uptake period following bilateral MCH CA1v injection. The MCH-treated group compared to the vehicle-treated group showed broad statistically significant differences in rCGU across the whole brain, indicative of changes in functional brain activation (Fig. 6a; Supplementary Table 1). Whereas the overall pattern of changes was bilateral, in some areas hemispheric lateralization was noted.

At the cortical level (including the cortical plate and subplate as defined by ref. [39]), increases in rCGU were observed in the MCH group in many association areas, including the anterior cingulate, prelimbic, infralimbic, retrosplenial, temporal association, ectorhinal, right orbital, right agranular insular, and left parietal areas. In the hippocampal formation, the CA1 (dorsal and ventral, with the CA1v being the targeted area for MCH injection), subiculum, entorhinal areas, left dentate gyrus, and left dorsal CA3 had increased rCGU, as did some sensory areas, including the auditory, visual, piriform area, right anterior olfactory nucleus, and right tenia tecta. CA1v MCH induced decreases in rCGU in the primary and supplemental somatosensory, visceral, left primary and secondary motor, left posterior, left gustatory, and left posterior agranular insular area as well as in the amygdalar nuclei of the cortical subplate, including basolateral, basomedial, cortical, and left lateral nucleus.

At the subcortical level, MCH induced increases in rCGU in the ACB, thalamus, left ZI, brain stem areas (superior colliculus, interpeduncular nucleus, superior central nucleus raphe, left dorsal periaqueductal gray, right parabrachial nucleus, right periolivary nuclei), and cerebellar central lobule. MCH induced decreases in rCGU in the striatum, lateral septal nucleus, left olfactory tubercle, amygdala (anterior area, central nucleus, medial nucleus), hypothalamus (dorsomedial nucleus, LHA, left anterior area), thalamus (mediodorsal nucleus, left ventral medial nucleus, left ventral anterior–lateral complex), and brain stem areas (central linear nucleus raphe; rostral linear nucleus raphe; ventral tegmental area; left external nucleus of inferior colliculus; right oculomotor nucleus, III; right red nucleus; right retrorubral area of midbrain reticular nucleus).

To determine functional connectivity between the MCH injection site (CA1v) and other brain regions, correlation analyses were conducted using the left CA1v as seed region of interest (ROI), as described[40]. Among the regions that showed functional connectivity (Supplementary Table 2), the connectivity analyses revealed statistically significant, positive correlation between the CA1v and ACB in the MCH group ($r$ = 0.76, $P$ = 0.02 at the ROI level) but not in the vehicle group ($r$ = 0.09, $P$ = 0.83 at the ROI level) (Fig. 6b), with a trend of difference in correlation coefficient between the groups ($P$ = 0.14, Fisher's $Z$-transform).

Based on these 2-DG functional mapping studies and functional connectivity analyses, as well as previous literature linking the ACB with impulsive behavior[11,41,42], we hypothesized that the ACB is a second-order target of CA1v-projecting MCH neurons. To further investigate this pathway, animals received injection of FG into the ACB shell, combined with IHC and FISH analyses to identify MCH terminals and MCHR1 mRNA, respectively, in the CA1v. Injection sites were based on our

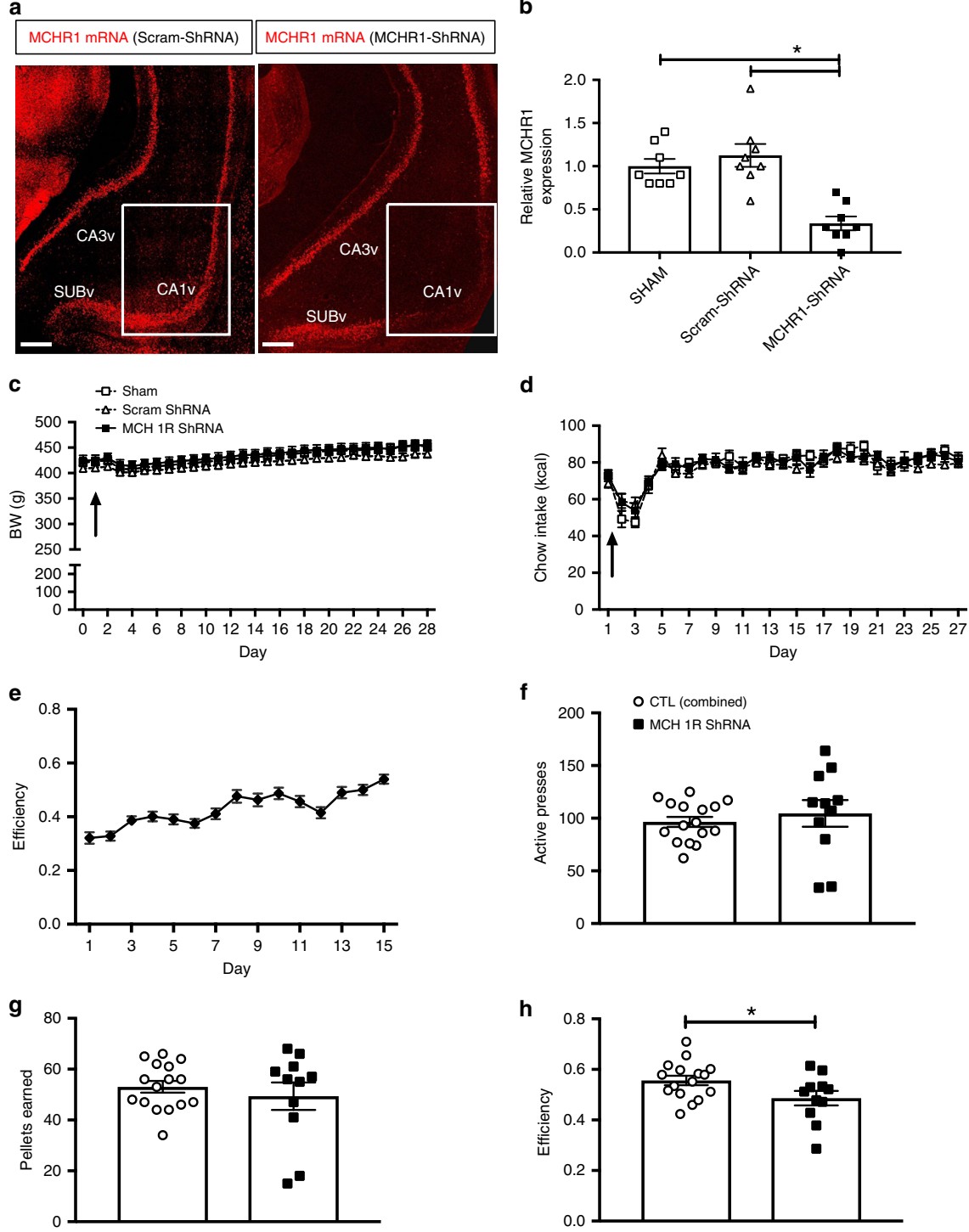

**Fig. 5** MCHR1 knockdown increases food impulsivity. **a** Representative images showing MCHR1 gene expression via fluorescence in situ hybridization and **b** relative MCHR1 gene expression (relative to Sham controls) via qPCR in CA1v in scrambled control shRNA (Scram-shRNA) or MCHR1 shRNA injected animals (one-way ANOVA with Tukey's multiple comparisons test; $n = 8$/group; $P < 0.05$); scale = 200 μm. **c** Body weight and **d** chow intake before (day 0) and after surgery in animals injected with either Scram-shRNA ($n = 8$), MCHR1 shRNA ($n = 11$), or Sham injected ($n = 8$); arrow indicates surgery date. **e** Efficiency data during the training phase in DRL 20 (before RNAi AAV injections). **f–h**; Student's two-tailed, unpaired $t$ test; $n = 11, 16$ (combined controls) **f** Number of active lever presses, **g** number of pellets earned and **h** efficiency in the DRL task following RNAi AAV incubation ($P = 0.04$). Data shown as mean ± SEM; *$P < 0.05$. Source data are provided as a source data file

functional connectivity analyses and also on prior reports suggesting that, while the ACB shell is implicated in both impulsive action and impulsive choice, the ACB core is more associated with motivational and motor aspects of impulsivity[43], which were not affected by altered MCH signaling in the vHP.

Results revealed back-labeled cells in the CA1v that [1] contain gene expression for the MCHR1 (~88% of back-labeled cells contained MCHR1) and [2] are in close apposition MCH terminals (Fig. 7a, b). These data suggest that the ACB may be a second-order target for CA1v-projecting MCH neurons.

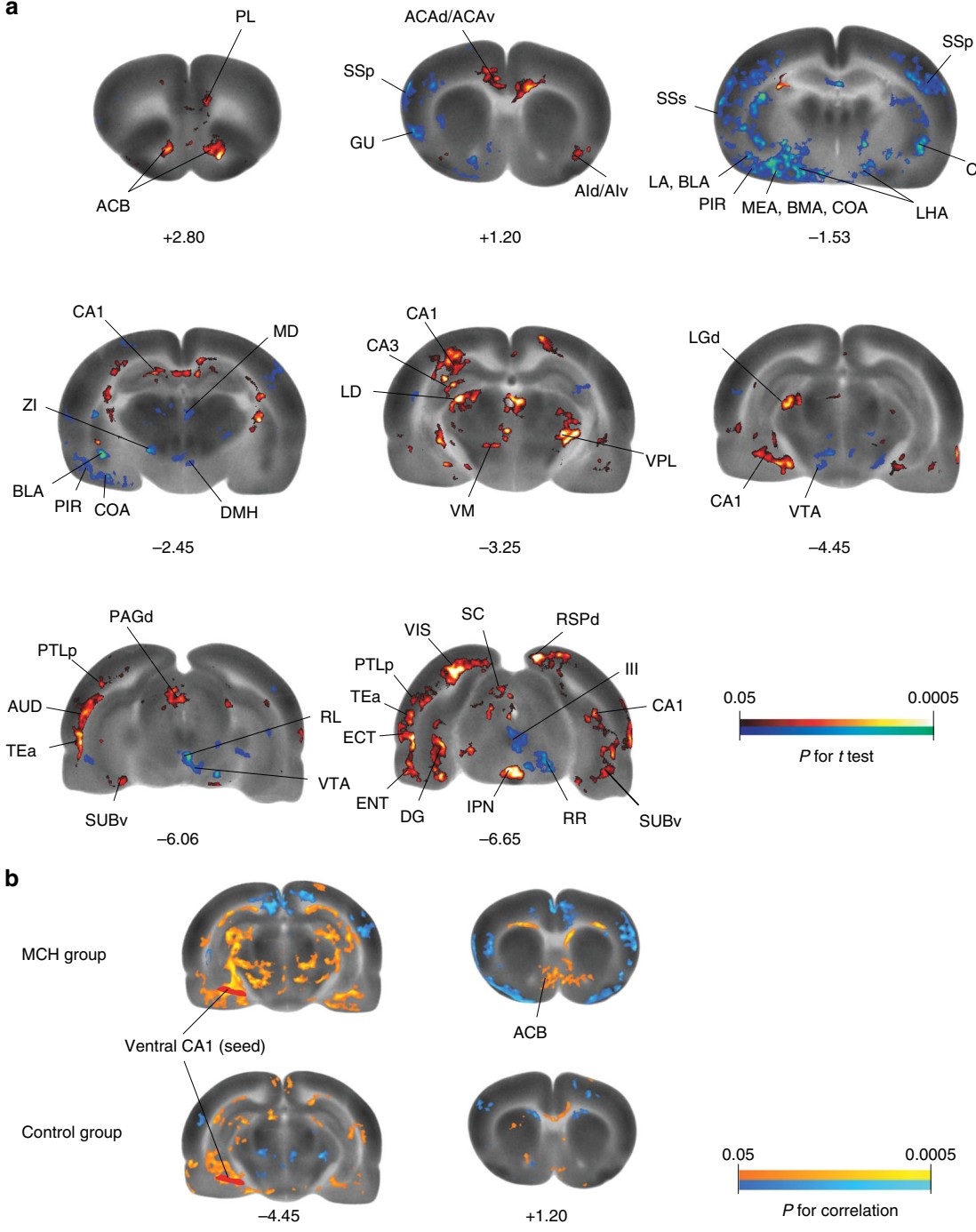

**Fig. 6** Neural responses following vHP MCHR1 activation. **a** Color-coded overlays over a selection of representative coronal sections of the template brain show significant MCH-induced changes in regional cerebral glucose uptake (rCGU) as an indirect measurement of brain activation (Student's *t* test, red/blue: increase/decrease in rCGU in the MCH group compared to the control group; $n = 9$ and 8, respectively). Numbers under images are approximate level along the anterior–posterior axis relative to the bregma (in mm). **b** Seed correlation analysis revealed statistically significant positive correlation between the seed in the ventral CA1 and ACB shell area in the MCH but not in the control group (Pearson's correlation, orange/blue: positive/negative correlation with the seed). ACB nucleus accumbens, ACAd/ACAv anterior cingulate area, dorsal/ventral part, AId/AIv agranular insular area, dorsal/ventral part, AUD auditory area, TEa temporal association areas, BLA basolateral amygdalar nucleus, BMA basomedial amygdalar nucleus, CA1/CA3 CA1/CA3 area of the HP, COA cortical amygdalar nucleus, CP caudoputamen, DG dentate gyrus, DMH dorsomedial hypothalamic nucleus, ECT ectorhinal area, ENT entorhinal area, GU gustatory area, III oculomotor nucleus, IPN interpeduncular nucleus, LA lateral amygdalar nucleus, LD lateral dorsal nucleus thalamus, LGd lateral geniculate complex, dorsal part, LHA lateral hypothalamic area, MD mediodorsal nucleus thalamus, MEA medial amygdalar nucleus, PAGd periaqueductal gray, dorsal division, PIR piriform area, PL prelimbic area, PTLp parietal region, posterior association areas, RL rostral linear nucleus raphe, RR midbrain reticular nucleus, retrorubral area, RSPd retrosplenial area, dorsal part, SC superior colliculus, SSp primary somatosensory area, SSs supplemental somatosensory area, SUBv subiculum, ventral part, VIS visual areas, VM ventral medial nucleus thalamus, VPL ventral posterolateral nucleus thalamus, VTA ventral tegmental area, ZI zona incerta). Abbreviations are taken from ref. [39]

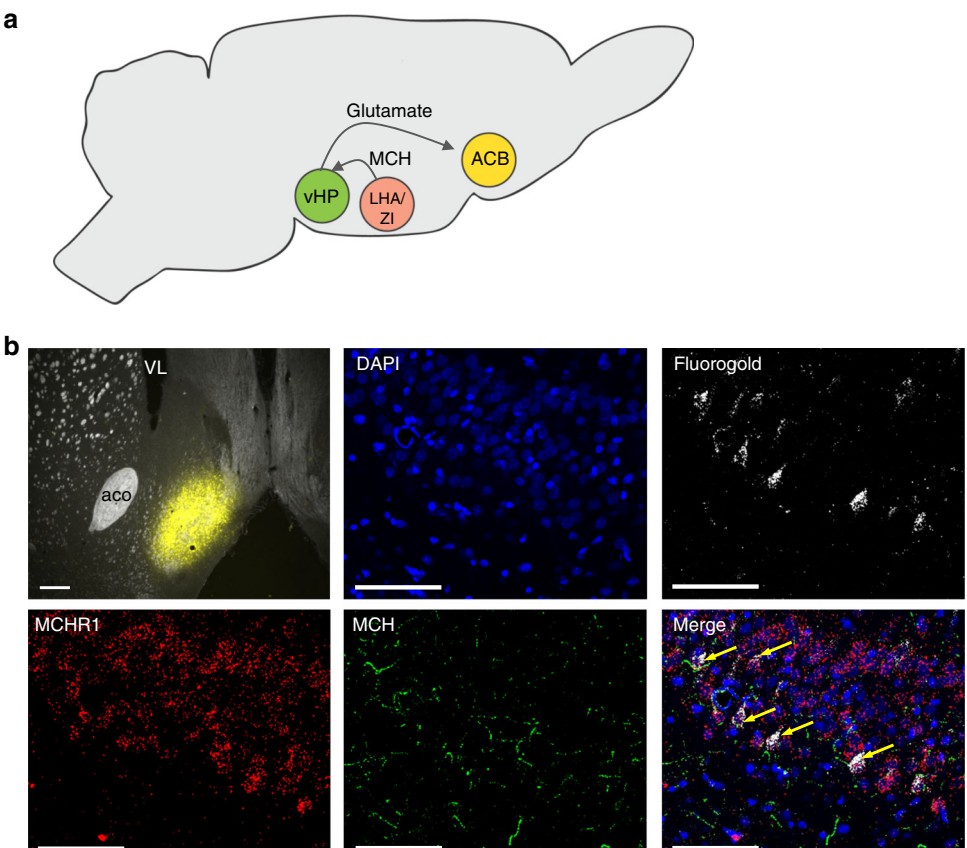

**Fig. 7** The ACB is a target for vHP-projecting MCH neurons. **a** A cartoon schematic showing the projection pathway from MCH neurons in the lateral hypothalamus/zona incerta (LHA/ZI) to the vHP and then to the ACB. **b** (top left) Injection site for fluorogold in the ACB in a 30-µm coronal section taken from a rat brain at 1.1 mm anterior to bregma; scale = 200 µm. Retrogradely labeled cells are shown in the CA1 subregion of the vHP in a 30-µm coronal section taken from a rat brain at 4.7 mm posterior to bregma. DAPI nuclear counterstain is shown in blue, retrogradely labeled fluorogold-positive cells are shown in white. Fluorescence in situ hybridization for the MCHR1 is shown in red and immunofluorescence staining for MCH is in green. Arrows point to MCH fiber appositions to MCHR1-positive cells containing fluorogold, indicating that these vHP neurons both receive input from MCH neurons and project to the ACB. Images taken at ×20 magnification; scale = 50 µm. ACB nucleus accumbens, aco anterior commissure, LHA lateral hypothalamus, vHP ventral hippocampus, VL lateral ventricle, ZI zona incerta

## Discussion

Our results identify a novel projection pathway involving MCH signaling from the LHA/ZI to the vHP that regulates behavioral impulsivity. These findings demonstrate a previously undiscovered mechanism through which MCH modulates behavior and further support emerging findings that the vHP is a critical brain substrate for modulating impulsivity and food-motivated behaviors[12,13,44]. Both chemogenetic and pharmacological activation of central nervous system (CNS) MCH signaling increased impulsivity in rats, an outcome that was recapitulated via vHP site-specific injection of MCH or by activating the specific MCH projections to the vHP (field CA1). As impulsivity is a common characteristic of metabolic disease and several prevalent neuropsychiatric and behavioral disorders, these findings have implications with regards to understanding the neurobiological underpinnings of physical and mental health.

Impulsive behavior has several underlying contributing factors, and therefore we sought to determine which of these factors were affected by vHP MCH signaling. For example, in addition to reduced inhibitory control and impaired knowledge of and/or appreciation of action consequences, impulsive behavior may reflect greater incentive motivation to obtain a reward. Indeed, our present and prior results reveal that central MCH signaling (via MCH ICV injections or chemogenetic activation of MCH neurons, indiscriminate of projection pathway) increases food

consumption[29]. However, increased incentive salience for food reward is unlikely to account for MCH vHP-mediated impulsivity, as elevated vHP MCH signaling had no effect on free feeding home cage consumption or motivated operant responding for palatable high-fat/high-sugar food. It is also possible that vHP MCH-driven impulsivity is secondary to altered physical activity levels, a possibility supported by data showing that both global MCH knockout[34–36] and pharmacological MCHR1 blockade[37] are associated with hyperactivity. However, elevated vHP MCH signaling had no effect on either lever pressing for the inactive lever or on activity levels in the open field test, suggesting that altered activity levels are unlikely to be causally related to the observed impulsivity. Finally, an additional alternative underlying factor that may be driving the impulsive behavior observed following activation of the MCH-vHP pathway is impaired clock speed timing, as the vHP plays a role in timing[45] and timing accuracy is involved with both of our rodent model tests of impulsivity (DRL, delay discounting). However, this is also unlikely as chemogenetic activation of MCH neurons had no effect on the animals' ability to accurately time 20 s (the critical time interval for the DRL impulsivity task) in a classic rodent timing task (peak interval). Together, these results suggest that the pathway from MCH neurons in the LHA/ZI to the vHP promotes impulsive behavior rooted in something other than heightened reward value, altered physical activity, or impaired

timing and is more likely based on a combination of impaired inhibitory control and reduced consideration of action consequences.

We reasoned that, if MCHR1 activation in the vHP increases impulsive behavior, chronic MCHR1 knockdown would reduce impulsive behavior. Surprisingly, our results showed that animals behaved more impulsively when MCHR1 levels were knocked down in the vHP, indicating an increase of impulsivity when vHP MCHR1 tone is perturbed in either direction. Further corroborating these findings, we also observed similar results when the gene precursor for the MCH peptide was targeted for knockdown in MCH neurons that project to the vHP. Similar bi-directional outcomes have been previously reported with the MCHR1 system. For example, MCH overexpression in the brain increases food intake[46], yet MCHR1 whole-brain knockdown has also been shown to promote hyperphagia[47,48], an effect that may be secondary to increased arousal and activity. One possible explanation for these and the present findings is that gain-of-function approaches targeting the MCH system (e.g., site-targeted pharmacology, chemogenetics) could potentially yield over-compensatory MCHR1 desensitization, or similarly, chronic loss-of-function approaches (e.g., genetic knockdown) could perhaps lead to overcompensatory MCR1 hypersensitization. Thus a useful area for follow-up examination is to investigate MCHR1 sensitivity in response to both acute and chronic manipulations that effect MCH ligand or receptor availability. Overall, however, our data identify a critical role for the MCH system in regulating impulsive behavior and join with other findings that CNS G protein-coupled receptor signaling systems can paradoxically yield similar behavioral outcomes when signaling is either upregulated or downregulated[49,50].

For our latter shRNA experiment, targeted RNAi of MCH in vHP-projecting LHA neurons resulted in a nonsignificant ~7% knockdown of global MCH mRNA. While this outcome is consistent with our anatomical data using FG and CAV-2-Cre approaches estimating that between 5% and 15% of all MCH neurons project to the vHP, the lack of significant group differences in global MCH expression, while not surprising, should be noted as a limitation of this experiment.

Chemogenetic activation of vHP-projecting MCH neurons may be engaging additional brain regions through branching collateral projections of these neurons. Indeed, while MCH neurons have extensive projections throughout the neuroaxis, very little is understood about the collateral targets of specific MCH projection pathways. Here we utilized a dual-viral neural pathway tracing approach to label axon collaterals of LHA neurons that project to the vHP. By combining this with MCH IHC, results identified the BLA as a collateral target of vHP-projecting MCH neurons. While these findings highlight the need for further investigation into the behavioral relevance of MCH neuron combined signaling to the vHP and BLA, MCH signaling to the BLA is unlikely to be mediating the observed increased impulsivity following vHP-projecting neuron activation. For example, pharmacological MCH administration to the vHP, which does not directly influence BLA MCH signaling, produced a similar increase in impulsivity in the DRL task compared to DREADD-mediated activation of vHP-projecting MCH neurons.

To identify candidate brain regions recruited downstream of vHP MCHR1 signaling, we analyzed regional cerebral glucose uptake (rCGU) following vHP MCHR1 activation in awake rats at rest. Notably, MCHR1 activation in the vHP altered glucose utilization in brain regions linked with response inhibition and impulsivity in humans. For example, we observed significant alterations bilaterally in the anterior cingulate, the right orbito-frontal cortex, medial prefrontal (prelimbic, infralimbic) areas, striatum, and motor regions (Fig. 6a, Supplementary Table 1). In humans, functional magnetic resonance imaging data during response inhibition has been shown to involve bilateral anterior cingulate cortex, insula, right orbitofrontal cortex, right dorso-lateral prefrontal cortex, and right supplementary motor areas[51]. The striatum and the HP, particularly the dorsal and ventral CA1, also had altered patterns of glucose utilization in our study, and these regions have been linked to impulsivity in humans[52,53].

Results from functional connectivity analyses revealed that activity in the vHP drug target site (CA1 subregion) is functionally connected with the ACB (Fig. 6b). Complemented by pathway tracing neuroanatomical analyses, these functional connectivity results suggest a putative polysynaptic neuronal circuit whereby vHP neurons (predominantly glutamatergic pyramidal neurons) receiving input from MCH neurons in the LHA/ZI project to the ACB. Previous findings link the ACB with impulsive control[11,41,42]; however, the behavioral relevance of the vHP-to-ACB direct projection pathway in mediating impulsive behavior requires further investigation.

Based on the established role of the central MCH system in promoting food intake and body weight gain in rodents[19–22], this system is currently a target for obesity pharmacotherapy development[54,55]. Obesity has been associated with both impulsive action[6] and impulsive choice[7], and related, impulsivity is also correlated with excessive food intake and weight gain[2,3,5,56]. The present study implicates a role for the MCH system in regulating impulsivity, and moreover, these effects were specific to impulsive behavior and were independent of alterations in hunger, food motivation, locomotor activity, or clock speed timing. Thus these data broaden the known potential of the MCH system for obesity therapeutics to include possibilities for treating other neuropsychological disorders with impulsive behavior as a characteristic (e.g., excessive gambling, drug abuse).

## Methods

**Animals**. Male Sprague Dawley rats (Envigo, Indianapolis, IN, USA) weighing 300–400 g were individually housed in wire-hanging cages in a climate controlled (22–24 °C) environment with a 12:12 h light/dark cycle. Except where noted, rats were given ad libitum access to water and standard rodent chow (LabDiet 5001, LabDiet, St. Louis, MO). Experiments were performed in accordance with NIH Guidelines for the Care and Use of Laboratory Animals, and all procedures were approved by the Institutional Animal Care and Use Committee of the University of Southern California.

**Stereotaxic cannula implantation**. Rats were first anesthetized with an intraperitoneal injection of ketamine (90 mg/kg)/xylazine (2.8 mg/kg)/acepromazine (0.72 mg/kg), prepped for surgery, and placed in a stereotaxic apparatus. For ICV cannulae, unilateral guide canulae (26-gauge, Plastics One) were surgically implanted targeting the lateral ventricle using the following coordinates[57]: −0.9 mm anteioror/posterior (AP), +1.8 mm medial/lateral (ML), −2.6 mm dorsal/ventral (DV) (0 reference point for AP and ML at bregma, 0 reference point for DV at skull surface at target site). Placement for the lateral ventricle cannula was verified by elevation of cytoglucopenia resulting from an injection of 210 μg (2 μl) of 5-thio-D-glucose (5tg)[58] using an injector that extended 2 mm beyond the end of the guide cannula. A postinjection elevation of at least 100% of baseline glycemia was required for subject inclusion. Animals that did not pass the 5tg test were retested with an injector that extended 2.5 mm beyond the end of the guide cannula and, upon passing 5tg, were subsequently injected using a 2.5-mm injector instead of a 2-mm injector for the remainder of the study.

For targeting the vHP, bilateral cannulae (26-gauge, Plastics One, Roanoke, VA) were implanted at the following stereotaxic coordinates[57]: −4.9 mm AP, +/−4.8 mm ML, −6.1 mm DV. Injectors for drug administration projected 2 mm beyond the guide cannulae. Placements for vHP cannulae were verified postmortem by injection of blue dye (100 nl, 2% Chicago sky blue ink) through the guide cannulae. Data from animals with dye confined to the vHP were included in the analyses.

**Immunohistochemistry**. Rats were anesthetized and sedated with a ketamine (90 mg/kg)/xylazine (2.8 mg/kg)/acepromazine (0.72 mg/kg) cocktail, then transcardially perfused with 0.9% sterile saline (pH 7.4) followed by 4% paraformaldehyde (PFA) in 0.1 M borate buffer (pH 9.5; PFA). Brains were dissected out and post-fixed in PFA with 15% sucrose for 24 h, then flash frozen in isopentane cooled in dry ice. Brains were sectioned to 30-μm thickness on a freezing

microtome. Sections were collected in 5 series and stored in antifreeze solution at −20 °C until further processing.

General fluorescence IHC labeling procedures were performed[29]. The following antibodies and dilutions were used: rabbit anti-MCH (1:1000; Phoenix Pharmaceuticals, Burlingame, CA, USA), rabbit anti-RFP (1:2000, Rockland Inc., Limerick, PA, USA), and Guinea Pig anti FG (1:5000; Protos Biotech Corp., New York, NY, USA). Antibodies were prepared in 0.02 M potassium phosphate-buffered saline (KPBS) solution containing 0.2% bovine serum albumin and 0.3% Triton X-100 at 4 °C overnight. After thorough washing with 0.02 M KPBS, sections were incubated in secondary antibody solution. All secondary antibodies were obtained from Jackson Immunoresearch and used at 1:500 dilution at 4 °C, with overnight incubations (Jackson Immunoresearch; West Grove, PA, USA). Sections were mounted and coverslipped using 50% glycerol in 0.02 M KPBS and the edges were sealed with clear nail polish.

IHC detection of MCH was performed according to the following sequence (overnight incubations on a motorized rotating platform at 4 °C): Sections were removed from anti-freeze and washed in [1] 0.02 M KPBS (6 changes in 2 h), [2] KPBS with 0.3% hydrogen peroxide (15 min), [3] KPBS (3 changes), [4] KPBS with 0.3% Triton X-100 (45 min), [5] KPBS (3 changes), [6] KPBS with 2% donkey serum (10 min), [7] KPBS with 1% donkey serum, 0.1% Triton X-100, and rabbit anti-MCH antibodies [1:2000; rabbit anti-MCH][17]. Primary antibody incubation length was ~60 h. [8] KPBS (8 changes in 2 h), [9] KPBS with 0.1% Triton X-100 and biotinylated secondary antibodies (1:1000; biotinylated donkey anti-rabbit, Jackson Immunoresearch; overnight). [10] KPBS (6 changes), [11] KPBS with tertiary reagent (1:1000 of reagent A and B from ABC Elite kit, Vector Labs; 4 h), [12] KPBS (3 changes), [13] KPBS with 0.05% 3,3′-diaminobenzidine (Sigma) and 0.005% $H_2O_2$ (15 min), [14] KPBS (4 changes). Sections were then mounted, air-dried, dehydrated with ascending concentrations of alcohol solutions, cleared in xylene, and coverslipped with DePeX mounting medium.

Photomicrographs were acquired using either a Nikon 80i (Nikon DS-QI1,1280X1024 resolution, 1.45 megapixel) under epifluorescence or darkfield illumination or as optical slices using a Zeiss LSM 700 UGRB Confocal System (controlled by Zeiss Zen software).

**Intracranial virus injections.** For stereotaxic injections of viruses and tracers, rats were first anesthetized and sedated with a ketamine (90 mg/kg)/xylazine (2.8 mg/kg)/acepromazine (0.72 mg/kg) cocktail. Animals were shaved, surgical site was prepped with iodine and ethanol swabs, and animals were placed in a stereotaxic apparatus for stereotaxic injections. Viruses were delivered using a microinfusion pump (Harvard Apparatus, Cambridge, MA, USA) connected to a 33-gauge microsyringe injector attached to a PE20 catheter and Hamilton syringe. Flow rate was calibrated and set to 5 μl/min; injection volume was 200 nl/site. Injectors were left in place for 2 min postinjection. Following injections, animals were either sutured or surgically implanted with a cannula where described. All experimental procedures occurred 21 days post virus injection to allow for transduction and expression. Successful virally mediated transduction was confirmed postmortem in all animals via IHC staining using immunofluorescence-based antibody amplification to enhance the fluorescence followed by manual quantification under epifluorescence illumination using a Nikon 80i (Nikon DS-QI1,1280X1024 resolution, 1.45 megapixel).

**Designer receptors exclusively activated by designer drugs.** Bilateral stereotaxic injections of AAV2-rMCHp-hM3D(Gq)-mCherry or AAV2-DIO-rMCHp-hM3D(Gq)-mCherry were made at the following coordinates[57]: injection (1) −2.6 mm AP, ±1.8 mm ML, −8.0 DV; (2) −2.6 mm AP, ±1.0 mm ML, −8.0 DV; (3) −2.9 mm AP, ±1.1 mm ML, −8.8 DV; (4) −2.9 mm AP, ±1.6 mm ML, −8.8 DV (from the skull surface at bregma).

For the dual-virus approach for selective expression of DREADDs in MCH neurons that project to the vHP, bilateral injections of the canine adenovirus 2 Cre (CAV2 CRE; 200 nl) were delivered using the following coordinates[57]: −4.9 mm AP, 4.8 mm ML, −7.8 mm DV (from the skull surface at the injection site) prior to injections with the MCH AAV2-DIO-rMCHp-hM3D(Gq)-mCherry.

**Characterization of DREADD expression.** DREADD expression was quantified in 1 out of 5 series of brain tissue sections from the perfused brains cut at 30 μm on a freezing microtome based on counts for the fluorescence reporter mCherry. Immunofluorescence staining for red fluorescent protein (RFP) was conducted as described above to amplify the mCherry signal. Counts were performed in sections from Swanson Brain Atlas level 27–32[39], which encompasses all MCH-containing neurons. Cell counts were performed in all DREADD virus-injected animals. For MCH DREADD experiments, animals were excluded from all experimental analyses if fewer than 2/3 of the total number of MCH neurons were transduced with RFP (based on IHC staining for MCH). For the dual-virus cre-dependent MCH DREADD experiments, all animals were included in the experimental analyses. Counts were performed by 2 researchers using epifluorescence illumination using a Nikon 80i (Nikon DS-QI1,1280X024 resolution, 1.45 megapixel) and the average of the 2 counts was taken. Researchers who performed the counting were kept consistent between cohorts and blind to experimental assignments.

**AAV-mediated RNA interference for MCHR1.** A custom shRNA targeting MCHR1 mRNA was cloned and packaged into an AAV1 under the control of a U6 promoter and co-expressing GFP (AAV1-GFP-U6-r-MCHR1-shRNA; Vector Biolabs, Malvern, PA, USA). For screening, 4–5 shRNA candidates were transfected into HEK293 cells to compare the knockdown efficiency for each shRNA. A reporter assay was used to assess the knockdown efficiency of each shRNA candidates, and the best shRNA was used to make the AAV virus. The shRNA was validated in vitro for ~95% knockdown of the mRNA for PMCH. This virus is now commercially available from Vector Biolabs (Malvern, PA, USA) upon request. The sequence is as follows:

5′-CACC GGAGTGTCTCCTACATCAACAC TCGAG TGTTGATGTAGGAGACACTCC-TTTTT-3′

The targeting sequence is GGAGTGTCTCCTACATCAACA and the hairpin loop sequence is CTCGAG. A separate AAV1 containing shRNA targeting a scrambled nonsensical sequence along with GFP was used as a control. AAVs or artificial cerebrospinal fluid (aCSF) were delivered bilaterally to the vHP (AP: −4.9, ML: +/−4.8, DV: −7.8, with DV zero at the skull at the injection site) at an injection volume of 200 nl via pressure injection using the stereotaxic procedures described above. The titer for the MCHR1 shRNA was $3.3 \times 10^{13}$ GC/ml and for the scrambled shRNA $1.7 \times 10^{13}$ GC/ml.

**Neural pathway tracing.** For retrograde pathway tracing, rats were anesthetized and sedated with a ketamine (90 mg/kg)/xylazine (2.8 mg/kg)/acepromazine (0.72 mg/kg) cocktail and placed in a stereotaxic apparatus. Rats received a unilateral iontophoretic injection of FG (Fluorochrome LLC; 2% in 0.9% NaCl) targeting either vHP (n = 4): −4.7 mm AP, 4.6 mm ML, −6.4 mm DV (from the dura at the injection site) or the ACB (n = 4): 1.2 mm AP, 1.0 mm ML, −6.75 mm DV (from the dura at the injection site) (coordinates from ref. [57]). Iontophoresis was performed using a precision current source (Digital Midgard Precision Current Source, Stoelting, Wood Dale, IL, USA) as described previously[14]. Following a 12-day survival period, animals were fixation-perfused and tissue was harvested and processed for immunofluorescence.

**Fluorescence in situ hybridization.** Tissue sections were obtained as in Immunofluorescence and mounted on subbed glass slides (Fisher brand Superfrost Plus, Fisher Scientific, Hampton, NH, USA) and desiccated overnight (~16 h) in a vacuum desiccant chamber. Following 1 h and 45 min postfix in 4% PFA, sections were washed 5 × 5 min in KPBS and incubated for 30 min at 37 °C in a solution of 100 mM Tris (pH 8), 50 mM EDTA (pH 8), and 0.1% Proteinase K (10 mg/ml, Sigma P2308), then rinsed for 3 min in the same Tris and EDTA solution without Proteinase K. Sections were washed 3 min in a solution of 100 mM triethanolamine (pH 8) in water and then incubated for 10 min at room temperature with 0.25% acetic anhydride in 100 mM triethanolamine, then washed 2 × 2 min in 10% 20× saline-sodium citrate buffer. Prior to hybridization, sections were dehydrated in increasing concentrations of ethanol (50%, 70%, 95%, 100%, 100%). Sections were incubated with probes for 3 h (MCHR1, ACD Cat #413191; vGLUT1, ACD Cat #317001; GAD2, ACD Cat #435801). Reagents from the RNAscope® Fluorescent Multiplex Detection Reagent Kit v2 (Advanced Cell Diagnostics, Newark, CA, USA; Cat #: 323100) were used to amplify the probe as per the kit's instructions. Slides were coverslipped using ProLong® Gold Antifade Reagent (Cell Signaling, Danvers, MA, USA; Cat #: 9071s). Photomicrographs were acquired using a Nikon 80i (Nikon DS- QI1,1280X1024 resolution, 1.45 megapixel) under epifluorescent illumination using the Nikon Elements BR software.

**Drug preparation and intracranial pharmacological injection.** For ICV and vHP injections of MCH, MCH (Bachem Americas, Torrance, CA, USC; Cat #: H-2218.1000) was dissolved in aCSF and diluted to 0.5 μg/μl (for vHP injections) or 5 μg/μl (for ICV injection), except where noted. For chemogenetic activation of MCH neurons, CNO (National Institute of Mental Health; 18 mmoles in 2 μl) or 33% dimethyl sulfoxide (DMSO) in aCSF (daCSF; vehicle control in 2 μl) was administered ICV. Animals were handled and habituated to injections prior to testing. All injections were delivered through a 33-gauge micro-syringe injector attached to a PE20 catheter and Hamilton syringe. For vHP injections, a micro-infusion pump (Harvard Apparatus) was used. The flow rate was set to 5 μl/min and 100 nl injection volume. Injectors were left in place for 30 s to allow for complete infusion of the drug. For ICV injections, 1 μl (MCH and aCSF) or 2 μl (CNO and daCSF) was delivered by manually plunging the Hamilton syringe.

**Differential reinforcement of low rates of responding.** The protocol that we used for the differential reinforcement of low rates of responding task (DRL) was modified from refs. [59–61]. Rats were first habituated in their home cage to 45 mg pellets containing 35% kcal from fat with sucrose (F05989, Bio-Serv, Frenchtown, NJ, USA). On each day of DRL training, chow was removed from the home cage 1 h prior to training, which began at the onset of the dark cycle, and chow was returned to the animals following DRL training. For DRL, animals were placed in an operant chamber (Med Associates, Fairfax, VT, USA) containing an active (reinforced with 1 palatable 45 mg pellet, with stimulus light activated during a reinforced lever press) and inactive (non-reinforced) lever. The task is 45 min long (one session per treatment), during which time the levers are extended, and the

animals may respond as many times as they choose for the entire 45 min. For the first 5 days of training, animals were on a DRL0 schedule, where each active lever press is reinforced with a 0-s time delay. Animals were then switched to a DRL5 schedule for 5 days, where rats must withhold pressing for a 5-s interval after pressing for each subsequent lever press to be reinforced, followed by 5 days of DRL10 (10 s withholding period) and 10 days of DRL 20 (20 s withholding period). A 2-day rest period occurred in between each 5 days of training. Efficiency in the DRL task was calculated as the number of pellets earned/the number of active lever presses.

On test days, food was removed 2 h prior to the lights going off and behavioral testing began at lights offset. For investigating the effects of MCH injection on DRL, animals were randomized to receive either aCSF or MCH using a counterbalanced within-subjects design, with 72 h between treatments 1 and 2. Injections were given 45 min prior to behavioral testing. For the effects of chemogenetic activation of MCH neurons on performance in DRL, animals were randomized to receive either daCSF or CNO using a counterbalanced within-subjects design with 72 h between treatments 1 and 2. CNO or daCSF were given 1.5 h prior to behavioral testing.

**Food intake studies**. A separate cohort of rats was used for each feeding experiment with food intake analyses occurring in the animal's home cage. For both studies using laboratory chow (LabDiet 5001,13% fat, 29% protein, 58% carbohydrate by kcal, LabDiet, St. Louis, MO, USA) and high-fat diet (Research Diets D12451, 45% fat, 20% protein, 35% carbohydrate by kcal, Research Diets Inc., New Brunswick, NJ, USA), home cage food was removed 2 h prior to the lights going off. Animals were randomized to receive either aCSF, 0.5 μg, or 1 μg MCH (for vHP studies) or aCSF, 5 μg, or 10 μg MCH (for ICV studies) in a counterbalanced within-subjects design. Injections were given 45 min prior to the lights going off and pre-weighed amounts of the test diet were deposited in the home cage immediately after the lights went out. For chemogenetic activation of MCH neurons, rats were randomized to receive either CNO (National Institute of Mental Health; 18 mmol in 2 μl) or 33% DMSO in aCSF (daCSF; vehicle control in 2 μl) ICV. Injections were given 1.5 h prior to the lights going off and pre-weighed amounts of the test diet were deposited in the home cage immediately after the lights went out. Spill papers were placed underneath the cages to collect food crumbs. Food spillage was weighed and added to the difference between the initial hopper weight and the hopper weight at each measurement time point. A total of 72 h was allotted between treatments.

**Delay discounting task**. The protocol for the delay discounting task was modified from refs. [33,62]. Rats were first habituated to palatable 45 mg sucrose pellets (F0023, Bio-Serv, Frenchtown, NJ, USA) in their home cage, then trained using a fixed ratio-1 (FR1) schedule to lever press for palatable 45 mg sucrose pellets in an operant chamber (Med Associates). On each day of the initial training, only the right or the left lever were retracted (in a counterbalanced fashion) and training continued until the animal reached the criterion of 50 lever presses in 45 min on both levers. Animals were then trained in a series of 4 blocks of 5 choice trials, where one lever consistently delivered 1 pellet per lever press, and the other lever delivered 4 pellets per press in a time delay of 0, 15, 30, and 45 s corresponding to block 1, 2, 3, and 4. Each block began with a forced trial on each lever, in random order. A stimulus light above each lever was lit during the forced trial and during the choice trials when the levers were extended. For each choice trial, levers remained extended for 10 s or until a lever response was made, after which the levers were retracted and the stimulus light remained illuminated over the pressed lever until pellets were dispensed. Preference for the smaller immediate reward over the larger, delayed reward is indicative of delay discounting.

On test day, food was removed 2 h prior to the lights going off and behavioral testing began at lights offset. Animals were randomized to receive either aCSF or MCH using a counterbalanced within-subjects design, with a washout period of 72 h between treatments 1 and 2. Injections were given 45 min prior to behavioral testing, and food was returned after the testing was complete.

**Open field test**. The apparatus used for the open field test is an opaque gray plastic bin (60 cm × 56 cm), which was positioned on a flat table in an isolated room with a camera directly above the center of the apparatus. Desk lamps were positioned to deliver indirect light on all corners of the maze such that the lighting in the box measured 30 lux in all corners and in the center. At the start of the 10-min test, each animal is placed in the open field apparatus in the same corner facing the center of the maze. All sessions were video recorded and ANY-maze video tracking software (Stoelting, Wood Dale, IL) was used for activity tracking. Total distance traveled was measured by tracking movement from the center of the rat's body.

On test days, food was removed 2 h prior to the lights going off and behavioral testing began when the lights turned off. For investigating the effects of MCH injection on open field activity levels, animals were randomized to receive either aCSF or MCH using a counterbalanced within-subjects design, with a washout period of 72 h between treatments 1 and 2. Injections were given 45 min prior to behavioral testing.

**Fixed interval (FI) and peak interval task**. Following recovery from surgery, rats (n = 6) were habituated for 15 min in their home cage to 20% sucrose solution. Rats were then acclimated to the conditioning boxes for two sessions where they received a total of 32, 0.1 ml sucrose deliveries to a recessed food cup, presented on a variable time (VT) 240 s schedule. Subsequently they received 10 sessions of FI training. In these sessions, during each trial (presented on a VT 60 s schedule) a lever was extended and the house light was illuminated. Each response to the lever was reinforced on a FI 20 s schedule resulting in sucrose delivery and occurred contemporaneously with both the retraction of the lever and the termination of the house light. Each session was completed after either 120 min had elapsed or 50 reinforcers had been delivered. During each of the remaining 20 sessions, rats received random presentation of 25 FI 20 s trials and 25 unreinforced probe trials. In each probe trial, the lever and house light were presented for 60 s followed by an added random period of time with a mean of 20 s driven by a Gaussian distribution. At the termination of the trial, the lever and the house light were retracted and extinguished, respectively. For the final four sessions, rats received 2 μl ICV infusions of either 18 mmol CNO or 0.2 M PBS, 15 min prior to the start of the session. Data from these test days were combined, resulting in 50 peak interval probe trials under chemogenetic stimulation via CNO and 50 trials under control PBS conditions.

**PR task**. The PR protocol was modified from ref. [63]. Training occurred in operant conditioning boxes (Med Associates; Fairfax, VT, USA) in 1 h sessions over 6 days. An FR1 with autoshaping procedure was used during the initial 2 days, where each lever press was reinforced with a 45-mg pellet (35% kcal from fat enriched with sucrose, F05989, Bio-Serv, Frenchtown, NJ). For the autoshaping component, a pellet was dispensed every 600 s that elapsed without operant-based reinforcement. The animals then received 2 days of FR1 schedule with no autoshaping, followed by 2 days of FR3 training, where 3 presses were required for each pellet earned. For all procedures, there was an active (reinforced) and an inactive (non-reinforced) lever. On test day, the response requirement of the PR schedule increased progressively as previously described[63–65] using the following formula: $F(i) = 5e^{0.2i} - 5$, where $F(i)$ is the number of lever presses required for next pellet at $i$ = pellet number. The breakpoint for each animal was defined as the final completed lever press requirement that preceded a 20-min period without earning a reinforcer.

On test day, food was removed 2 h prior to the lights going off and behavioral testing began at lights offset. Animals were randomized to receive either aCSF or MCH using a counterbalanced within-subjects design, with a washout period of 72 h between treatments 1 and 2. Injections were given 45 min prior to behavioral testing, and food was returned after the testing was complete.

**Quantitative PCR (qPCR) quantification of in vivo MCHR1 knockdown**. Brains from sham, scramble, or MCHR1-injected animals were rapidly removed and flash frozen in isopentane and stored at −80 °C. Serial coronal sections (50 μm) of the midbrain/forebrain were mounted on a slide and viewed immediately under a fluorescent microscope (Nikon 80i) until EGFP-expressing neurons were visualized. Total RNA was extracted from 1 mm micropunches of the vHP CA1 region using the Purelink RNA Mini kit (ThermoFisher Scientific, Canoga Park, CA, USA). cDNA was synthesized from 125 ng total RNA using the iScript cDNA Synthesis Kit (BioRad Inc., Hercules, CA, USA) and qPCR was carried out in duplicate using TaqMan Fast Advanced Master Mix (ThermoFisher Scientific). Samples were run using StepOne Plus Real Time System with the primer/probe sets for MCHR1 (Rn00755896_m1) and GAPDH (internal control; Rn0177563_g1) (ThermoFisher Scientific). Relative mRNA expression was calculating using the comparative Ct method.

**Functional brain mapping**. Autoradiographic 2-DG uptake metabolic mapping was used to access functional brain activation following bilateral injection of MCH into the vHP. Animals were implanted bilaterally with guide cannula for drug injection targeting the ventral CA1 as described above and allowed to recover for at least 3 weeks before the mapping experiment. Animals were individually housed and randomly assigned into one of the two groups: MCH (n = 9, body weight 411 ± 20 g on the day of mapping) and vehicle (n = 8, body weight 414 ± 20 g). For 3 days prior to the mapping experiment, animals were acclimated each day to handling (5 min), the experiment room (15 min), and a cylindrical, plexiglass experimental arena (30 min, diameter = 30 cm, height = 30 cm) under low-level ambient lighting.

Relative rCGU was measured according to the method of Sokoloff[66] with modification[67]. On the day of mapping, food was removed 120 min before the injection of 2-DG to prevent interference with glucose uptake by food consumption. Seventy minutes after food removal, the animals were injected bilaterally with MCH solution (5 μg/μl in aCSF) or vehicle (aCSF) through an injection cannulae projected 2 mm beyond the guide cannulae, at a 5 μl/min rate and 100 nl injection volume per hemisphere. The animal was left in its home cage for 50 min before receiving an IP injection of 2-DG (Moravek Inc., Brea, CA, USA; cat # MC355, 0.1 μCi/g body weight in 0.53 ml saline). The animal was immediately placed inside the arena and left there for 45 min for 2-DG uptake mapping in a no task, resting state. Thereafter, the animal was deeply anesthetized with 4%

isoflurane and euthanized by an intracardiac injection of 3 M potassium chloride solution (1 ml).

Brains were extracted, flash frozen in methylbutane on dry ice, and serially sectioned (57 coronal 20-μm slices, 300-μm interslice distance beginning at ~4.5 mm anterior to the bregma) in a cryostat (HM550, Microm International GmbH, Walldorf, Germany). Slices were heat-dried on glass slides and exposed to Kodak Biomax MR films (Eastman Kodak, Rochester, NY, USA) for 3 days at room temperature. Autoradiographic images of brain slices were digitized on an 8-bit gray scale using a voltage-stabilized light box (Northern Lights Illuminator, Interfocus Imaging Ltd., Cambridge, UK) and a Retiga 4000R charge-coupled device monochrome camera (Qimaging, Surrey, Canada). For each animal, a three-dimensional brain was reconstructed from 57 digitized autoradiograms (voxel size: $40 \times 300 \times 40$ μm$^3$) in ImageJ. Adjacent sections were aligned manually in Photoshop (version 9.0, Adobe Systems Inc., San Jose, CA, USA) and using TurboReg, an automated pixel-based registration algorithm implemented in ImageJ (version 1.35, http://rsbweb.nih.gov/ij/).

We and others have adapted the statistical parametric mapping (SPM) package (Wellcome Centre for Neuroimaging, University College London, London, UK) for the analysis of rodent autoradiographic cerebral blood flow[68] and CGU data[69]. For preprocessing, one artifact free brain was selected as reference. All brains were spatially normalized to the reference brain in SPM (version 5). Spatial normalization consisted of applying a 12-parameter affine transformation followed by a nonlinear spatial normalization using three-dimensional discrete cosine transforms. All normalized brains were then averaged to create a final rat brain template. Each original brain was then spatially normalized to the template. Final normalized brains were smoothed with a Gaussian kernel (full-width at half-maximum = $240 \times 300 \times 240$ μm$^3$) to improve the signal-to-noise ratio. Proportional scaling was used to normalize whole-brain average CGU across animals.

Unbiased, voxel-by-voxel Student's $t$ tests between the MCH and vehicle groups were performed across the whole brain to access changes in rCGU following MCH injection into the ventral CA1. Threshold for significance was set at $P < 0.05$ at the voxel level and an extent threshold of 200 contiguous voxels. This combination reflected a balanced approach to control both Type I and Type II errors. The minimum cluster criterion was applied to avoid basing our results on significance at a single or small number of suprathreshold voxels. Brain regions were identified according to a rat brain atlas[39].

Seed ROI correlation analysis was used to assess functional connectivity of the ventral CA1[40]. Structural ROI of the ventral CA1 was hand drawn in MRIcro (version 1.40, http://cnl.web.arizona.edu/mricro.htm) over the template brain in the left hemisphere according to the rat brain atlas. The structural ROI was then intersected with clusters showing MCH-induced significant increases in rCGU to create a functional seed ROI. Mean optical density of the seed ROI was extracted for each animal using the MarsBaR toolbox for SPM (version 0.42, http://marsbar.sourceforge.net/). Correlation analysis was performed in SPM for each group using the seed values as a covariate. Threshold for significance was set at $P < 0.05$ at the voxel level and an extent threshold of 200 contiguous voxels. Regions showing significant correlations in rCGU with the seed ROI are considered functionally connected with the seed. Statistical significance of between-group differences in correlation coefficients (between the seed and another ROI) was evaluated using Fisher's $Z$-transform test ($P < 0.05$).

**Statistical analyses**. Analysis of variance (ANOVA) were performed using the GraphPad Prism 7.0 Software (GraphPad Software Inc., San Diego, CA, USA) and $t$ tests were performed using either GraphPad Prism 7.0 software or Microsoft Excel for Mac (v. 15.26; Microsoft Inc., Redmond, WA, USA). DRL, PR, and shRNA quantification data were analyzed using a Student's two-tailed paired $t$ test. All comparisons of the effects of either CNO or MCH on chow or high-fat diet intake were analyzed using a repeated-measures ANOVA with Tukey's post hoc test for multiple comparisons. Delay discounting data were analyzed using a two-way repeated-measures ANOVA with Sidak test for multiple comparisons. For all statistical tests, the $\alpha$ level for significance was 0.05.

**Reporting summary**. Further information on research design is available in the Nature Research Reporting Summary linked to this article.

## Data availability

All data generated and analyzed for this manuscript are available from the corresponding author (S.E.K.) upon reasonable request. The source data underlying Figs. 1b–e, g–j, 3a–d, g–j, 4a–i, and 5b–h; Supplemental Figs. 1, 2a–d, 4a, b, and 7a–g; and all inactive lever press data as well as the qPCR data for the pMCH RNAi experiment are provided as a Source Data file. The data from this manuscript are available in the Open Science Framework Repository https://doi.org/10.17605/OSF.IO/4XEVQ.

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

## Acknowledgements

The research was supported by DK118402 and DK104897 and to S.E.K., and DK107333 to T.M.H., DK116558 to A.N.S., DK118944 to C.M.L., DK105155 to M.R.H., DK111475 to A.W.J., and DK118000 to E.E.N. Clozapine-*N*-Oxide was kindly provided by the National Institute of Mental Health. Rae Lan, Jamie Clarke, Lekha Chirala, and Ryan Fatemi are acknowledged for their critical contributions to the research.

## Author contributions

E.E.N., Z.W., D.P.H. and S.E.K. designed the experiments. Neuroanatomical experiments and analysis were performed by E.E.N., J.D.H., A.M.C. and S.E.K. Virogenetic injections and behavioral experiments were performed by E.E.N., T.M.H., A.-H.J., L.M.R., A.N.S., L.T., C.M.L., S.J.T., L.A.S. and E.A.D. For the shRNA, tissue analyses and qPCR were performed by L.P.S., E.A.D. and M.R.H. Cerebral metabolic mapping experiments and analyses were performed by Z.W., D.P.H. and E.E.N. Cav2CRE was provided by M.D. The paper was written by E.E.N., Z.W. and S.E.K. with additional editorial input from the other authors.

## Competing interests

M.R.H. has received research support from investigator-initiated sponsored proposals from Novo Nordisk, Zealand Pharma, and Boehringer-Ingelheim. The other authors declare no competing interests.
