## [Peer Review File · Nature Communications]

Reviewers' Comments:

Reviewer #1:

Remarks to the Author:

In this study, the authors report that LH MCH to ventral hippocampus project regulates impulsivity, but not eating. This is an interesting dissociation of functions of LH MCH cells. A good range of appropriate tools are used. Overall, however, there are some key points that are not investigated which leave the conclusions of the study somewhat unclear and vague:

1) Despite many experiments, it is not clear from this study whether MCH signals increase or reduce impulsivity. The reported data is self-contradictory and this is not adequately investigated.

Specifically:

It is shown in Fig 5H that reducing MCH function (by MCH receptor inhibition) reduces task efficiency, but in preceding figures it is shown that increasing MCH function (with dREADDs or MCH infusion) also reduces task efficiency. The critical issue of the sign of the effect of MCH signalling on impulsivity thus remains unclear from this study. The authors briefly mention in Discussion that above findings mean that there might be an inverted-U shaped relationship between MCH tone and impulsivity, but this claim is not adequately assessed. Inverted-U would be interesting, but to demonstrate it the authors need to vary the intensity of one/same type of manipulation (e.g. MCH concentration, optogenetic or chemogenetic stimulation intensity, etc) on the x-axis, while measuring impulsivity on the y axis. This evidence is missing from the paper, thus overall there is not enough evidence to conclude whether MCH signals increase, reduce, or have an inverse-U relation with, impulsivity.

2) MCH signalling is known to be "anti-locomotive" (eg Shimada et al Nature 1998), and thus its inhibition/activation may make animals more/less likely to move. Please explain more clearly how such a general effect can be distinguished from impulsivity?

3) Fig. 7: The authors make a conceptual leap here by proposing a novel disinhibitory circuit that accounts for the effects in the paper (LH-MCH  vHP  ACB). However, the evidence they present for this (immuno and tracing in fig 7) is at best preliminary. Functional or anatomical connections of MCH-modulated cells in vHP to ACB are not demonstrated by these histological experiments. The model shown in A is thus purely speculative at this stage and this should be adequately reflected in the presentation.

Reviewer #2:

Remarks to the Author:

This manuscript by Noble et al. addresses the neural circuitry and signals underlying impulsivity, a maladaptive behavior intrinsic to some psychiatric illnesses and eating styles. Using a variety of approaches to modulate central melanin concentrating hormone (MCH) levels, MCH neuronal activity and MCHR1 expression, the authors identify the contribution of MCH neurons projecting to the ventral hippocampus (vHP) and signalling herein on active lever responding in several operant tasks of impulsive food seeking.

First, the authors show that ICV injection of MCH increased active lever responding in a reinforcement task assessing capacity to wait for the food pellet (DRL task) without affecting the number of pellets earned, thus the reinforcing effects of food. Chemogenetic activation of MCH neurons is shown to have similar actions. They then confirm that a subset of MCH neurons project to the CA1 subregion and characterize MCHR1 neurons within as mostly GLUT neurons using RNA scope. Both injection of MCH

in CA1 and chemogenetic activation of MCH neurons that project to the vHP (Cav2-cre + custom DREADD AAV expressing Gq under MCH promoter) are shown to have similar actions in DRL task and in delayed discounting task (for intra vHP MCH injections) but not on feeding, food motivation (PR task) or capacity to accurately time reinforcement delivery. These additional tests convincingly demonstrate effects are not secondary to changes in appetite or pace-making mechanisms. In an unexpected manner, MCHR1 knockdown in the vHP had similar actions to increases active lever responding, a finding that is hard to explain without further analyses. Finally using glucose utilization mapping (2-DG autoradiography) and connectivity analyses following local MCH injections, the authors find that the nucleus accumbens (ACB) is a downstream target of MCHR1 vHP neurons.

The authors employ a slew of elegant and complementary molecular and pharmacological approaches to demonstrate importance of not only MCH neurons projecting to vHP but also actions of peptide itself in the vHP on food-based responding in well-validated tests of impulse control. The results are very interesting and novel. Experimental design and statistical analyses appear solid. My major concern is the lack of control experiments evaluating changes in locomotion, which could well contribute to the operant endpoints and thus correct interpretation of results. The authors do not present inactive lever responses, which is a common readout to include in such tests. If inactive responses are similarly increased this could suggest an effect of manipulations to stimulate arousal and/or hyperactivity (rather than impulsivity), processes that the MCH system is well implicated in as the authors point out. These data need to be presented throughout, plus comprehensive measures of locomotor activity in activity chambers for at least a subset of interventions performed.

My other major concern is interpretation of MCHR1 knockdown findings. These findings are puzzling. Suggesting a U function would require some data in the curve of the U. With the 67% KD in MCHR1 described one might expect there would be a smaller effect, as opposed to higher KD rates. The authors might be able to at least correlate KD % with behavioral readouts for each rat to support the U shaped function idea.

Other comments

1. More information is needed about how the DRL task was carried out. Was it 1 session per Tx? An average of multiple sessions following Tx? Cited papers don't describe procedure used here.
2. It is important to see if the session times varied between groups. For example, in Fig 4, did MCHR1 KD rats achieve criteria number of pellets before the control group and thus have shorter session times? This could confound interpretation.
3. Why was CNO injected 1.5h prior to testing whereas MCH was injected 45 min before? Please explain. And as a comment: 33% DMSO concentration for CNO is unusually high. High DMSO concentrations injected centrally can be sedating.
4. It is not clear what the control group is in Fig. 5 F- H. The control should be the scramble shRNA but symbols suggest that it is the SHAM. And, do authors have any within subject data to support interpretation of these results? In other words, before vs after shRNA injection ?
5. Why were fluorogold injections in the ACB shell rather than core? Justify in text
6. The 2-DG analyses are very nice, but does MCH affect glucose metabolism such that it could bias mapping?
7. Line 593: DLR acronym incorrectly used
8. Line 605: I believe authors mean "levers remained extended.."

Reviewer #3:

Remarks to the Author:

The manuscript by Noble et al. addresses the important question of the neural mechanisms of impulsive behavior. The authors describe a novel mechanism via MCH, and therefore also provide a potentially important link to metabolic disorders. Specifically, the authors show that gain-of-function manipulations of MCH neuron projections to vHP (e.g., chemogenetic activation, pharmacology), and loss-of-function via knockdown of MCHR1 in vHP, both increase impulsive behavior. The authors further implicate vHP MCHR1 projections to NAc. Overall, the manuscript is clearly written, describes a potentially important novel mechanism, and many of the experiments are well-controlled. However, I do have some concerns (detailed below) that need to be addressed before it can be published in Nature Communications. First and foremost among those is the apparent discrepancy between gain and loss of function studies, which might pertain to the limitations of the experimental approaches used in this study, which in turn might undermine the strength of the conclusions that can be reached. See my detailed explanations below.

Major points:

1. The authors show that gain-of-function manipulations of MCH neuron projections to vHP increase impulsive behavior. These include ICV injection of MCH, chemogenetic activation of MCH neurons and of MCH->vHP neurons. The authors also show that loss-of-function via knockdown of MCHR1 in vHP increases impulsive behavior. While the authors do discuss this issue and attempt to provide an explanation for it, I found this explanation vague and unconvincing. The main problem here is that the gain/loss-of-function discrepancy could actually be revealing the limitations of the technical approaches used. Generally, it could be that MCH and/or MCH neurons are also acting via a different mechanism. Specifically:

- a. The MCHR1 shRNA could have off-target effects in vHP, affecting the results. While these are notoriously hard to pin down, the way people usually address this is by showing similar results between 2-3 different shRNAs. As far as I could tell, this was not done here.
- b. ICV infusion of MCH could be acting via other brain regions, not just vHP.
- c. MCH->vHP neurons (projection-specific DREADDs) could be acting via collaterals to other brain regions (collaterals were not assessed here).

I suggest that the authors perform additional experiments to address this issue. For example, the authors could do one or a few of the following:

- The authors could show similar results using at least one other MCHR1 shRNA to preclude off-target effects.
- Local infusion of MCH and of MCHR1 antagonist (in separate experiments) in the vHP (not ICV) to see if it reproduces the paradoxical effects. This would at least control for the ICV infusion acting on other brain regions to produce these effects.
- The authors could combine DREADD activation of MCH->vHP + with local infusion of an MCHR1 antagonist in the vHP (not ICV).
- The authors could combine DREADD activation of MCH->vHP with vHP MCHR1 shRNA to show that the shRNA blocks the effect.

2. Many results presented (histology, 2DG imaging) are seemingly quantified, but there is no additional information to allow the reader to assess this. Also, some additional histological analyses are missing. Specifically:

- a. For all histological analyses, the authors only mention overall percentages (e.g., overlap between two markers). The authors should explicitly mention number of animals used and also mean±sem across animals in the Results section.
- b. The authors should analyze MCH->vHP collaterals to other brain regions. This would help interpret

the MCH->vHP DREADD experiments, seeing as CNO was infused ICV.

c. The authors present an interesting analysis of the CA1v-NAc correlation. However, it is hard to evaluate this without more context comparing it to correlation values with other brain region (e.g., do these correlation range from -0.8 to 0.8 or are they all between 0.6 and 0.8?). The authors should add supplementary table (similar to the one they have already included) with correlations of CA1v with other brain regions.

d. The authors have a full figure describing CAV->NAc projections. This is an interesting putatively relevant mechanism. However, it is presented without any quantification of CAV->NAc overlap with MCHR1. This should be added.

Minor points:

1. Ref 9 is missing
2. Could the authors clarify how FISH analyses found that exc/inh overlap with MCHR1 sums up to 127% overlap? Do so many neurons express both GAD2 and vGLUT2?
3. Two home-cage feeding experiments are presented. Could the authors clarify if these are two different cohorts?
4. Sample sizes 2DG imaging are mentioned only in Methods. The authors should present these in the Results section to allow readers to assess the reliability of the data.
5. The last sentence of the Results should be phrased more modestly. Instead of "... the ACB *is* a second order target..." it should be "... the ACB *could be* a second order target..."

Reviewers' comments:

Reviewer #1 (Remarks to the Author):

In this study, the authors report that LH MCH to ventral hippocampus project regulates impulsivity, but not eating. This is an interesting dissociation of functions of LH MCH cells. A good range of appropriate tools are used. Overall, however, there are some key points that are not investigated which leave the conclusions of the study somewhat unclear and vague:

We thank the reviewer for highlighting the intrigue of our data, and the broad scope of tools used to address our hypotheses. In this revised version we have addressed all of the key issues raised, thus enhancing the strength of the conclusions that can be drawn from our data.

1) Despite many experiments, it is not clear from this study whether MCH signals increase or reduce impulsivity. The reported data is self-contradictory and this is not adequately investigated. Specifically: It is shown in Fig 5H that reducing MCH function (by MCH receptor inhibition) reduces task efficiency, but in preceding figures it is shown that increasing MCH function (with dreads or MCH infusion) also reduces task efficiency. The critical issue of the sign of the effect of MCH signalling on impulsivity thus remains unclear from this study. The authors briefly mention in Discussion that above findings mean that there might be an inverted-U shaped relationship between MCH tone and impulsivity, but this claim is not adequately assessed. Inverted-U would be interesting, but to demonstrate it the authors need to vary the intensity of one/same type of manipulation (e.g. MCH concentration, optogenetic or chemogenetic stimulation intensity, etc) on the x-axis, while measuring impulsivity on the y axis. This evidence is missing from the paper, thus overall there is not enough evidence to conclude whether MCH signals increase, reduce, or have an inverse-U relation with, impulsivity.

We agree that our findings indicating similar behavioral outcomes for gain and loss of vHP MCH signaling are surprising, but we respectfully disagree that the data are contradictory in nature. Rather, we think that the data speak to the complexity of endogenous neuropeptidergic and G protein-coupled receptor signaling systems, and we cite similar examples in the Discussion section. However, we agree with the reviewer that due to the somewhat surprising nature of the outcome, additional experiments utilizing different methods should be included to confirm the validity of our findings. Thus, we now include an additional experiment utilizing a novel projection-specific approach for RNA interference in which shRNAs targeting the pMCH gene are introduced into MCH neurons that project to the vHP. Results from this experiment were similar to our RNA interference approach for the vHP MCH1 receptor in the previous submission, in that RNA interference targeting MCH peptide expression in vHP-projecting MCH neurons also increased impulsive responding for food in the DRL task.

These new data both confirm and extend our findings that perturbing the MCH system in the vHP in either direction leads to a similar behavioral outcome of elevated impulsivity. We do, however, note one limitation with this new experiment. Consistent with our anatomical data estimating that between 5-15% of all MCH neurons project to the vHP, our qPCR results revealed a ~7% reduction in global MCH mRNA expression in the knockdown group compared to controls. However, this difference failed to reach statistical significance. This is not surprising given that our qPCR analyses are measuring global MCH expression, and only a small % of MCH neurons will be targeted by the AAV2-retro shRNA injections in the vHP. Nevertheless, we report that cell-based in vitro analyses confirm ~94% knockdown of MCH mRNA expression with the shRNA sequence, and further we provide anatomical confirmation in a representative animal of retrogradely transfected GFP reporter expression in LHA MCH neurons,

suggesting that our approach of targeting MCH neurons with shRNA was successful (Suppl. Fig. 6, 7). We directly acknowledge this limitation in the manuscript (See Methods lines 259-280, and Discussion lines 396-401).

We agree that additional data points would be required in order to adequately confirm the presence of a true U-shaped curve, and therefore we have adjusted our interpretation of the data throughout the manuscript such that our conclusions better reflect the data presented, which shows that perturbing the vHP MCHR1 system in either direction (by upregulating or downregulating signaling) causes a similar behavioral outcome (increased impulsivity).

2) MCH signalling is known to be "anti-locomotive" (eg Shimada et al Nature 1998), and thus its inhibition/activation may make animals more/less likely to move. Please explain more clearly how such a general effect can be distinguished from impulsivity?

We thank the reviewer for identifying this potential confound with regards to interpretation of our impulsivity behavioral outcomes, and thus we have now included new data and a new experiment investigating the effects of MCH in the vHP on locomotor activity. First, for each of the DRL tasks we now include lever pressing data on the inactive lever, which is an indicator of general activity while in the operant chambers during the task (means and SEM listed in Results Section).

Second, we now include results from two new experiments using the open field task, which measures general locomotor activity in an open arena. There were no differences in distance travelled in the open field task when MCH was injected into the vHP at doses sufficient to elevate impulsive responding (Figure 4F). Similarly, there were no differences in distances travelled during the open field task in animals with pMCH shRNA targeting vHP-projecting MCH neurons (Supplemental Fig 8G). Thus, while increasing MCH signaling generally reduces locomotor activity (e.g. Shimada et al), our new data show that site-specific injection of MCH into the vHP has no effect on locomotor activity, nor does reducing gene expression of pMCH in MCH neurons that project to the vHP.

3) Fig. 7: The authors make a conceptual leap here by proposing a novel disynaptic circuit that accounts for the effects in the paper (LH-MCH  vHP  ACB). However, the evidence they present for this (immuno and tracing in fig 7) is at best preliminary. Functional or anatomical connections of MCH-modulated cells in vHP to ACB are not demonstrated by these histological experiments. The model shown in A is thus purely speculative at this stage and this should be adequately reflected in the presentation.

We have softened the language throughout to better reflect the data presented within the manuscript, as suggested.

Reviewer #2 (Remarks to the Author):

This manuscript by Noble et al. addresses the neural circuitry and signals underlying impulsivity, a maladaptive behavior intrinsic to some psychiatric illnesses and eating styles. Using a variety of approaches to modulate central melanin concentrating hormone (MCH) levels, MCH neuronal activity and MCHR1 expression, the authors identify the contribution of MCH neurons projecting to the ventral hippocampus (vHP) and signalling herein on active lever responding in several operant tasks of impulsive food seeking.

First, the authors show that ICV injection of MCH increased active lever responding in a reinforcement task assessing capacity to wait for the food pellet (DRL task) without affecting the number of pellets earned, thus the reinforcing effects of food. Chemogenetic activation of MCH neurons is shown to have similar actions. They then confirm that a subset of MCH neurons project to the CA1 subregion and characterize MCHR1 neurons within as mostly GLUT neurons using RNA scope. Both injection of MCH in CA1 and chemogenetic activation of MCH neurons that project to the vHP (Cav2-cre + custom DREADD AAV expressing Gq under MCH promoter) are shown to have similar actions in DRL task and in delayed discounting task (for intra vHP MCH injections) but not on feeding, food motivation (PR task) or capacity to accurately time reinforcement delivery. These additional tests convincingly demonstrate effects are not secondary to changes in appetite or pace-making mechanisms. In an unexpected manner, MCHR1 knockdown in the vHP had similar actions to increases active lever responding, a finding that is hard to explain without further analyses. Finally using glucose utilization mapping (2-DG autoradiography) and connectivity analyses following local MCH injections, the authors find that the nucleus accumbens (ACB) is a downstream target of MCHR1 vHP neurons.

The authors employ a slew of elegant and complementary molecular and pharmacological approaches to demonstrate importance of not only MCH neurons projecting to vHP but also actions of peptide itself in the vHP on food-based responding in well-validated tests of impulse control. The results are very interesting and novel. Experimental design and statistical analyses appear solid.

We thank the reviewer for the careful and thorough review of our manuscript, for highlighting the intrigue and the novelty of our findings, and for the suggested improvements. Our responses are detailed below:

My major concern is the lack of control experiments evaluating changes in locomotion, which could well contribute to the operant endpoints and thus correct interpretation of results. The authors do not present inactive lever responses, which is a common readout to include in such tests. If inactive responses are similarly increased this could suggest an effect of manipulations to stimulate arousal and/or hyperactivity (rather than impulsivity), processes that the MCH system is well implicated in as the authors point out. These data need to be presented throughout, plus comprehensive measures of locomotor activity in activity chambers for at least a subset of interventions performed.

As described above in our response to Reviewer #1, we now include lever pressing data on the inactive lever for each of the DRL tasks, as you suggested (means and SEM given in Results Section). We did not observe differences in inactive lever presses for any of the DRL experiments presented in the manuscript. As you further suggest, we also performed a measurement of locomotor activity for a subset of conditions. Notably, MCH injected into the vHP at a dose that elevated impulsive responding for food had no effect on distance travelled in the open field task (Figure 4F). Similarly, there were no differences in distances travelled during the open field task in animals with pMCH shRNA knockdown in vHP-projecting MCH neurons (Supplemental Fig 8G).

My other major concern is interpretation of MCHR1 knockdown findings. These findings are puzzling. Suggesting a U function would require some data in the curve of the U. With the 67% KD in MCHR1 described one might expect there would be a smaller effect, as opposed to higher KD rates. The authors might be able to at least correlate KD % with behavioral readouts for each rat to support the U-shaped function idea.

Reviewer #1 had a similar concern and we also agree that further data are required to confirm the presence of a U-shaped curve. We now include new data corroborating that bidirectional manipulations of the MCH system in the vHP cause a similar behavioral outcome (please see our response to Reviewer 1 comment #1 for more details). We have altered the language throughout such that we are no longer concluding a “U shaped curve” as we agree that this conclusion would require additional data points along the curve, an extensive analysis that is well beyond the scope of this manuscript.

Other comments

1. More information is needed about how the DRL task was carried out. Was it 1 session per Tx? An average of multiple sessions following Tx? Cited papers don't describe procedure used here.

We apologize for the oversight and have now included the following methods description: “The task is 45 minutes long (one session per treatment), during which time the levers are extended, and the animals may respond on the lever as many times as they choose for the entire 45 minutes.”

2. It is important to see if the session times varied between groups. For example, in Fig 4, did MCHR1 KD rats achieve criteria number of pellets before the control group and thus have shorter session times? This could confound interpretation.

We believe that the reviewer meant Figure 5 as Figure 4 does not contain MCHR1 KD data. There is no criterion to be reached in the DRL test, as indicated in the revised methods and in the statement responding to query 1 above. In case the Reviewer was referring to a task in Figure 4, which contains delay discounting, progressive ratio, and peak interval timing task data, we would also highlight that none of these tasks are dependent on a criterion number of pellets earned. For delay discounting, there is no criterion number of pellets that need to be achieved before the task stops. For PR, the task ends when the animal has not responded on the lever for 20 minutes (and not when a criterion has been reached), and for the peak interval timing task responses are measured during probe trials where no reinforcers are given.

3. Why was CNO injected 1.5h prior to testing whereas MCH was injected 45 min before? Please explain. And as a comment: 33% DMSO concentration for CNO is unusually high. High DMSO concentrations injected centrally can be sedating.

Differences in the timing of the drug injections are based on the time that it took to observe a feeding effect when comparing MCH with CNO/DREADDs activation of MCH neurons from our previous studies. In rats, we observe significant increases in food intake at 2 hours post injection (food given at dark onset) when MCH is injected 45 min prior to the onset of the dark cycle (present paper), or when CNO is given to animals 1.5 hours prior to dark onset expressing excitatory DREADDs in MCH neurons (Noble et al., *Cell Metabolism*, 2018). We agree that the concentration of DMSO is quite high, however the solubility of CNO requires a solvent to keep it in solution. We note that in the control condition, animals were injected with vehicle which was 33% DMSO without the CNO.

4. It is not clear what the control group is in Fig. 5 F- H. The control should be the scramble shRNA but symbols suggest that it is the SHAM. And, do authors have any within subject data to support interpretation of these results? In other words, before vs after shRNA injection?

We apologize for the accidental repeated use of symbols. The symbols have been changed in Fig. 5B-D so that they are now different from those in Fig. 5F-H. Additionally, we have changed the legend for the CTL group in Fig. 5F-H such that it now says “CTL (combined)” to reflect that we combined the control groups because they were not different from each other in any of the parameters tested.

We calculated within-subject’s differences in efficiency scores (before vs after shRNA injection), and consistent with reported results, the statistical comparison showed that animals injected with MCHR1 shRNA had significant efficiency reductions following shRNA injections (vs. pre surgery) compared with CTLs. However, we chose to not include this analysis in the revision, since we matched the groups based on terminal training performance, and subsequently observed significant group differences during DRL testing. Therefore, we believe this analysis, while corroborative, is redundant with our reported significant results.

5. Why were fluorogold injections in the ACB shell rather than core? Justify in text We now say in the text that injection sites were based on our functional connectivity analyses and also on data suggesting that while the ACB shell is implicated in both impulsive action and impulsive choice, the ACB core affects more motivational and motor aspects of impulsivity (PMID 24810333), which were not affected by MCH signaling in the vHP.

6. The 2-DG analyses are very nice, but does MCH affect glucose metabolism such that it could bias mapping? We are not aware of any other studies investigating the effects of MCH on central glucose metabolism independent of the general effect that the peptide has on the activity and metabolism of the cell itself, which is what we are measuring with the 2DG approach. While we do think that this concept is an interesting idea, we could not find relevant data to cite and discuss this further.

7. Line 593: DLR acronym incorrectly used
Removed acronym

8. Line 605: I believe authors mean “levers remained extended..”
Changed to “levers remained extended”

Reviewer #3 (Remarks to the Author):

The manuscript by Noble et al. addresses the important question of the neural mechanisms of impulsive behavior. The authors describe a novel mechanism via MCH, and therefore also provide a potentially important link to metabolic disorders. Specifically, the authors show that gain-of-function manipulations of MCH neuron projections to vHP (e.g., chemogenetic activation, pharmacology), and loss-of-function via knockdown of MCHR1 in vHP, both increase impulsive behavior. The authors further implicate vHP MCHR1 projections to NAc. Overall, the manuscript is clearly written, describes a potentially important novel mechanism, and many of the experiments are well-controlled. However, I do have some concerns (detailed below) that need to be addressed before it can be published in Nature Communications. First and foremost among those is the apparent discrepancy between gain and loss of function studies, which might pertain to the limitations of the experimental approaches used in this study, which in turn might undermine the strength of the conclusions that can be reached. See my detailed explanations below.

We thank the reviewer for highlighting the novelty of our findings and for identifying areas where the paper could be improved. We have addressed the Reviewer’s concerns below:

Major points:

1. The authors show that gain-of-function manipulations of MCH neuron projections to vHP increase impulsive behavior. These include ICV injection of MCH, chemogenetic activation of MCH neurons and of MCH->vHP neurons. The authors also show that loss-of-function via knockdown of MCHR1 in vHP increases impulsive behavior. While the authors do discuss this issue and attempt to provide an explanation for it, I found this explanation vague and unconvincing. The main problem here is that the gain/loss-of-function discrepancy could actually be revealing the limitations of the technical approaches used. Generally, it could be that MCH and/or MCH neurons are also acting via a different mechanism. Specifically:

- a. The MCHR1 shRNA could have off-target effects in vHP, affecting the results. While these are notoriously hard to pin down, the way people usually address this is by showing similar results between 2-3 different shRNAs. As far as I could tell, this was not done here.
- b. ICV infusion of MCH could be acting via other brain regions, not just vHP.
- c. MCH->vHP neurons (projection-specific DREADDs) could be acting via collaterals to other brain regions (collaterals were not assessed here).

I suggest that the authors perform additional experiments to address this issue. For example, the authors could do one or a few of the following:

- The authors could show similar results using at least one other MCHR1 shRNA to preclude off-target effects.
- Local infusion of MCH and of MCHR1 antagonist (in separate experiments) in the vHP (not ICV) to see if it reproduces the paradoxical effects. This would at least control for the ICV infusion acting on other brain regions to produce these effects.
- The authors could combine DREADD activation of MCH->vHP + with local infusion of an MCHR1 antagonist in the vHP (not ICV).
- The authors could combine DREADD activation of MCH->vHP with vHP MCHR1 shRNA to show that the shRNA blocks the effect.

To address (a) above, we now include an additional experiment in which an shRNA targeting the gene for the precursor to MCH (pMCH) is targeted to MCH neurons that project to the vHP (see response to Reviewer #1 comment 1). These new results further strengthen our interpretation of results presented in the first submission.

To address (b), we would like to highlight that we did perform pharmacological experiments where MCH peptide was injected specifically in the vHP at a dose that was ineffective for feeding effects when injected ICV (see figure 3A-D).

To address (c), this is a very interesting and generative suggestion. As mentioned above, our DREADDs data are corroborated by pharmacological site-specific injections where MCH is directly injected into the vHP. Thus, while collateral projections could contribute to the effects seen when activating MCH neurons that project to the vHP, MCH delivery in the vHP is sufficient to increase impulsive behavior, and thus possible collateral targets are unlikely mediating these effects. However, we do agree that characterizing the collateral projections of the vHP-projecting MCH neurons would be very interesting and would provide novel targets for follow-up research. Thus, we now include a new experiment wherein we identify that collateral projections of vHP-projecting MCH neurons can be found in the basolateral amygdala (Supplemental Fig. 5). We did not observe collateral projections in any additional

regions. We have now added an entire paragraph in the Discussion to discuss these results and the putative role of collateral projections in mediating observed effects (See Methods lines 188-194, Discussion lines 401-414).

2. Many results presented (histology, 2DG imaging) are seemingly quantified, but there is no additional information to allow the reader to assess this. Also, some additional histological analyses are missing. Specifically:

a. For all histological analyses, the authors only mention overall percentages (e.g., overlap between two markers). The authors should explicitly mention number of animals used and also mean±sem across animals in the Results section.

As indicated in the methods section, cell counts were performed using 1 out of 5 series. Cell counts were performed in duplicate by two blinded researchers and the average of their two counts were taken. Given that we did not count the whole population of neurons, we feel that it is more appropriate to use percentages than cell count numbers, as the absolute count numbers may mislead readers to think that there are fewer total cells than there actually are (as one would have to refer back to the methods section to obtain information about cell counting methods).

b. The authors should analyze MCH->vHP collaterals to other brain regions. This would help interpret the MCH->vHP DREADD experiments, seeing as CNO was infused ICV.

As indicated above, we now include an additional experiment where we analyze collateral projections as suggested (see Supplemental Fig 5).

c. The authors present an interesting analysis of the CA1v-NAc correlation. However, it is hard to evaluate this without more context comparing it to correlation values with other brain region (e.g., do these correlation range from -0.8 to 0.8 or are they all between 0.6 and 0.8?). The authors should add supplementary table (similar to the one they have already included) with correlations of CA1v with other brain regions.

We now include a Supplementary Table (Supplementary Table 2) showing the functional correlations of CA1v with additional brain regions.

d. The authors have a full figure describing CAv->NAc projections. This is an interesting putatively relevant mechanism. However, it is presented without any quantification of CAv->NAc overlap with MCHR1. This should be added.

We have now added the quantification to the results section, as suggested (~88% of back-labeled cells contained MCHR1).

Minor points:

1. Ref 9 is missing

This error has now been fixed.

2. Could the authors clarify how FISH analyses found that exc/inh overlap with MCHR1 sums up to 127% overlap? Do so many neurons express both GAD2 and vGLUT2? We cannot confirm from our data whether there is overlap of GAD2 and vGLUT1 (we did not measure vGLUT2) in the vHP, as our FISH probe channel overlap precludes our capacity to simultaneously analyze MCHR1+GAD2+vGLUT2. However, emerging research suggests the existence of dual phenotype GABA/glutamate neurons in many brain regions, including the hippocampus (PMID 20519538, PMID 15927685, PMID 25749864). In fact, vGlut and vGAT have even been shown to coexist within vesicles, where the corelease of these transmitters may prevent excitotoxicity (PMID 20519538). We thank the reviewer for bringing to our attention an exciting avenue for future research.

3. Two home-cage feeding experiments are presented. Could the authors clarify if these are two different cohorts?

We now state that feeding experiments were conducted in different cohorts of rats in the methods section.

4. Sample sizes 2DG imaging are mentioned only in Methods. The authors should present these in the Results section to allow readers to assess the reliability of the data.

We apologize for the oversight and now include the n for the 2DG experiment in the Results section → figure legend for Fig. 6.

5. The last sentence of the Results should be phrased more modestly. Instead of "... the ACB *is* a second order target..." it should be "... the ACB *could be* a second order target..."

The last sentence now says, "These data suggest that the ACB may be a second-order target for CA1v-projecting MCH neurons."

Reviewers' Comments:

Reviewer #1:

Remarks to the Author:

I appreciate the efforts the authors made in revision. I respectfully disagree, however, with their argument that their data are not self-contradictory.

My reasoning is based on the following understanding of their observations:

Observation 1a: MCH downregulation in vHP-projecting neurons increases impulsivity.

Observation 1b: MCHR downregulation in vHP increases impulsivity

Thus, conclusion 1 = Natural MCH signalling in vHP reduces impulsivity.

Observation 2: MCH injection into vHP increases impulsivity.

From this, conclusion 2 = (Artificially-increased) MCH signalling in vHP increases impulsivity.

I hope the authors can now see that the above conclusions 1 and conclusion 2 directly contradict each other.

I find this very strange, and no adequate explanation is proposed. With all due respect: just referring to "system complexity" is not an adequate or useful scientific justification, and neither is their reference (in Rebuttal) to other published unexplained contradictions.

I hope the authors can acknowledge better the above contradiction, and discuss some possible/plausible scientific explanation. For example, artificial application of MCH at high doses might desensitise MCH receptors, for example leading to their internalisation and loss of function (there are many precedents of this in receptor pharmacology studies). So this artificial overstimulation experiment, intended as a gain-of-function, may actually in reality be a loss-of-function experiment. If the latter is true, there is no self-contradiction in the author's data. I hope to see some scientific arguments of this kind in the paper, so the field can move forward in knowledge while having full awareness of caveats of artificial stimulation.

Reviewer #2:

Remarks to the Author:

The authors have responded very well to concerns. Revisions, including addition of new data, significantly strengthen results and interpretation.

Reviewer #3:

Remarks to the Author:

The authors have adequately addressed all my concerns.

Reviewer #1 (Remarks to the Author):

I appreciate the efforts the authors made in revision. I respectfully disagree, however, with their argument that their data are not self-contradictory.

My reasoning is based on the following understanding of their observations:

Observation 1a: MCH downregulation in vHP-projecting neurons increases impulsivity.
Observation 1b: MCHR downregulation in vHP increases impulsivity
Thus, conclusion 1 = Natural MCH signalling in vHP reduces impulsivity.

Observation 2: MCH injection into vHP increases impulsivity.
From this, conclusion 2 = (Artificially-increased) MCH signalling in vHP increases impulsivity.

I hope the authors can now see that the above conclusions 1 and conclusion 2 directly contradict each other.

I find this very strange, and no adequate explanation is proposed. With all due respect: just referring to "system complexity" is not an adequate or useful scientific justification, and neither is their reference (in Rebuttal) to other published unexplained contradictions.

I hope the authors can acknowledge better the above contradiction, and discuss some possible/plausible scientific explanation. For example, artificial application of MCH at high doses might desensitise MCH receptors, for example leading to their internalisation and loss of function (there are many precedents of this in receptor pharmacology studies). So this artificial overstimulation experiment, intended as a gain-of-function, may actually in reality be a loss-of-function experiment. If the latter is true, there is no self-contradiction in the author's data. I hope to see some scientific arguments of this kind in the paper, so the field can move forward in knowledge while having full awareness of caveats of artificial stimulation.

In response to the reviewer's suggestions, in the 3rd paragraph of the Discussion we now have removed the word "complexity" regarding the MCH system. Further, we appreciate the useful suggestion about discussing MCH1R sensitization (and/or desensitization) as a possible mechanism for both the present results, as well as those cited within the same paragraph for previous studies showing hyperphagia following both gain- and loss-of-function approaches targeting the MCH system. Lastly, we have now added in this paragraph that the present results, coupled with these similar previous findings, highlight a need for future research to investigate MCH1R sensitization following both acute and chronic manipulations to the MCH system. The full updated paragraph is pasted below this response, with new text in yellow (revised manuscript file lines 382-402).

"We reasoned that if MCHR1 activation in the vHP increases impulsive behavior,

chronic MCHR1 knockdown would reduce impulsive behavior. Surprisingly, our results showed that animals behaved more impulsively when MCHR1 levels were knocked down in the vHP, indicating an increase of impulsivity when vHP MCHR1 tone is perturbed in either direction. Further corroborating these findings, we also observed similar results when the gene precursor for the MCH peptide was targeted for knockdown in MCH neurons that project to the vHP. Similar bi-directional outcomes have been previously reported with the MCHR1 system. For example, MCH overexpression in the brain increases food intake ⁴⁶, yet MCHR1 whole brain knockdown has also been shown to promote hyperphagia ^{47, 48}, an effect that may be secondary to increased arousal and activity. One possible explanation for these and the present findings is that gain-of-function approaches targeting the MCH system (e.g., site-targeted pharmacology, chemogenetics) can yield overcompensatory MCR1 desensitization, or similarly, that chronic loss-of-function approaches (e.g., genetic knockdown) can lead to overcompensatory MCR1 hypersensitization. Thus, a useful area for follow-up examination is to investigate MCHR1 sensitivity in response to both acute and chronic manipulations that effect MCH ligand or receptor availability. Overall, however, our data identify a critical role for the MCH system in regulating impulsive behavior, and join with other findings that CNS G protein-coupled receptor signaling systems can paradoxically yield similar behavioral outcomes when signaling is either up- or down-regulated ^{49, 50}.”

Reviewer #2 (Remarks to the Author):

The authors have responded very well to concerns. Revisions, including addition of new data, significantly strengthen results and interpretation.

Reviewer #3 (Remarks to the Author):

The authors have adequately addressed all my concerns.

Reviewers' Comments:

Reviewer #1:

Remarks to the Author:

The authors conclusion that MCH manipulations can modulate impulsivity in any direction is now adequately described, though importance of this for normal behavior remains unclear.

REVIEWERS' COMMENTS:

Reviewer #1 (Remarks to the Author):

The authors conclusion that MCH manipulations can modulate impulsivity in any direction is now adequately described, though importance of this for normal behavior remains unclear.

We thank the reviewer and agree that future studies will be important to elucidate the full significance of our findings to normal behavior.